# Cryo-EM structures of ryanodine receptors and diamide insecticides reveal the mechanisms of selectivity and resistance

Lianyun Lin[1,2,9], Changshi Wang [3,9], Wenlan Wang[1,2], Heng Jiang[1,2], Takashi Murayama [4], Takuya Kobayashi [4], Hadiatullah Hadiatullah[1,2], Yu Seby Chen [5], Shunfan Wu [6], Yiwen Wang [1], Henryk Korza[7], Yucheng Gu [7], Yan Zhang [1,2], Jiamu Du [3], Filip Van Petegem [4] ✉ & Zhiguang Yuchi [1,2,8] ✉

The resistance of pests to common insecticides is a global issue that threatens food production worldwide. Diamide insecticides target insect ryanodine receptors (RyRs), causing uncontrolled calcium release from the sarcoplasmic and endoplasmic reticulum. Despite their high potency and species selectivity, several resistance mutations have emerged. Using a chimeric RyR (chiRyR) approach and cryo-electron microscopy (cryo-EM), we investigate how insect RyRs engage two different diamide insecticides from separate families: flubendiamide, a phthalic acid derivative, and tetraniliprole, an anthranilic compound. Both compounds target the same site in the transmembrane region of the RyR, albeit with different poses, and promote channel opening through coupling with the pore-forming domain. To explore the resistance mechanisms, we also solve two cryo-EM structures of chiRyR carrying the two most common resistance mutations, I4790M and G4946E, both alone and in complex with the diamide insecticide chlorantraniliprole. The resistance mutations perturb the local structure, directly reducing the binding affinity and altering the binding pose. Our findings elucidate the mode of action of different diamide insecticides, reveal the molecular mechanism of resistance mutations, and provide important clues for the development of novel pesticides that can bypass the resistance mutations.

Although antibiotic resistance in pathogenic microorganisms is well known, pest resistance against insecticides has also become a major issue, threatening crop yields worldwide and causing an estimated annual economic loss of USD 10 billion in the US alone[1]. Excessive pesticide use, driven by resistance, can also negatively impact the environment and public health. Diamide insecticides, including phthalic acid and anthranilic diamides, are among the best-selling insecticides globally because of their excellent potency, low non-target toxicity, and lack of cross-resistance with existing insecticides[2–5]. Since their introduction to the market in the late

2000s, they have rapidly gained popularity, with annual sales exceeding USD 2 billion[6–8].

The molecular target of diamide insecticides is the ryanodine receptor (RyR), a large ion channel expressed in the membranes of the sarcoplasmic and endoplasmic reticulum (SR and ER). RyRs govern the release of $Ca^{2+}$ into the cytosol, driving multiple events, including muscle contraction[9,10]. Mammalian species encode three different RyR isoforms (RyR1-3), whereas insects have only one isoform. RyRs are named after ryanodine, a plant alkaloid with insecticidal properties[11,12]. Although ryanodine itself is no longer in commercial use and also

doesn't discriminate between humans and insects, RyRs have remained a target for insecticides. Diamide insecticides directly engage and activate insect RyRs, causing uncontrolled $Ca^{2+}$ release, leading to tonic muscle contraction, paralysis, feeding cessation, and eventually, death of the insect[2,13].

Several generations of diamide insecticides are in commercial use. The first, flubendiamide (FLU) (Fig. 1a), a phthalic acid derivative, was introduced in 2007 and has shown high activity against most major lepidopteran pests[2]. A year later, the first anthranilic diamide, chlorantraniliprole (CHL) (Fig. 1a), was introduced, resulting in lower toxicity and greater potency than those of FLU[5]. More recently, a third-generation anthranilic diamide, tetraniliprole (TET) (Fig. 1a), has become available for controlling a wide range of lepidopteran, coleopteran, dipteran, and hemipteran pests. Compared with traditional diamide insecticides, TET has a broader spectrum and a longer-lasting effect[14–16]. Despite their commercial success, the exact mechanisms of action, selectivity, and toxicity of different diamide insecticides have remained largely unclear, hindering the rational design of the next generation of RyR-targeting insecticides.

Owing to their extensive use, many pests, including the diamondback moth (*Plutella xylostella*), fall armyworm (*Spodoptera frugiperda*), rice stem borer (*Chilo suppressalis*), and tomato leafminer (*Tuta absoluta*), have developed resistance to diamide insecticides. This resistance extends to both phthalic acid and anthranilic diamide insecticides[13,17–19]. The emergence of resistant populations was first observed in the diamondback moth (DBM) in Thailand and the Philippines, where the RyR mutation G4946E results in a >200-fold decrease in sensitivity to CHL and FLU[20]. Independently, three new mutations (E1338D, Q4594L, and I4790M) were found in the DBMs from the Chinese Yunnan Province, decreasing the sensitivity to CHL > 2000-fold[21]. The I4790M (DBM numbering) was also found in the Brazilian *Spodoptera frugiperda* population, resulting in 225-fold and 5,400-fold decreased sensitivity to CHL and FLU, respectively[17]. Other mutations (I4790K, Y4667D/C, Y4891F, and G4946V, DBM numbering) have been identified in different insect populations and are directly linked to resistance[22–25]. To date, I4790M and G4946E (DBM numbering) are the most common mutations found in multiple insect species, leading to high-level resistance. These mutations have also been observed in combination, further exacerbating resistance[21,25].

At very high concentrations, diamide insecticides can also engage mammalian RyRs, allowing us to identify a binding site for CHL in the transmembrane region of rabbit RyR1 (rRyR1) via cryo-EM[26]. However, diamide insecticides exhibit ~100-1,000 times greater selectivity for insect RyRs than for mammalian RyRs, and the binding site, located in the pseudo-voltage sensing domain (pVSD), is not conserved between insects and mammals, with approximately 49.5% sequence identity (Supplementary Fig. 1). In this study, we produced a chimeric RyR (chiRyR) that fully preserves the diamide-binding pocket from the fall armyworm *Spodoptera frugiperda* RyR (*Sf* RyR), a major lepidopteran agricultural pest. We solved several structures in complex with different classes of diamide insecticides, including RyRs carrying resistance mutations, providing insights into the mechanisms of selectivity and resistance against diamide insecticides.

## Results

### Design of a chimeric RyR that recapitulates diamide insecticide binding

Although mammalian RyRs can be obtained through recombinant expression and FKBP-based affinity purification[27,28], insect RyRs have proven to be more challenging for larger-scale production and affinity purification because of their lower affinity for FKBP, which has enabled efficient purification. The overall sequence identity between *Sf* RyR and rRyR1 is 45.2%, with higher identity (~60%) in their transmembrane regions, where the diamide binding site is located (Supplementary

Fig. 1). Therefore, we designed a series of insect-like chimeric RyRs that contain the diamide-binding pocket from *Sf* RyR and the remaining parts from rabbit RyR1 (rRyR1).

Both phthalic acid and anthranilic diamide insecticides, represented by FLU and TET respectively, are highly selective against insect RyRs compared with mammalian RyRs. To determine their potencies against different RyR constructs, we utilized a fluorescence-based assay that monitors ER $Ca^{2+}$ levels in RyR-expressing HEK293 cells[29]. The addition of diamide insecticides results in $Ca^{2+}$ release and a concomitant decrease in ER $Ca^{2+}$ levels. We found that the $EC_{50}$ values of FLU and TET against rRyR1 were 4.5 μM and 7.6 μM, respectively. In contrast, the $EC_{50}$ values against *Sf* RyR were significantly lower at 59.3 nM and 174.5 nM, respectively (Fig. 1b and Supplementary Table 1), confirming their high selectivity against the insect RyR[30–32].

To investigate the impact of specific residues on diamide binding, we mutated four rRyR1 residues previously found to contact the diamide CHL in our rRyR1-CHL complex (PDB ID 6M2W) (Fig. 1c, d) into the corresponding *Sf* RyR residues. The individual mutations R4563K, F4564Y, C4657I, and L4792S (rRyR1 numbering) significantly decreased the $EC_{50}$ values for both FLU and CHL, with the largest decrease observed with the L4792S mutation (Supplementary Table 2). The combination of all four mutations fully restored the high-affinity binding of both FLU and TET, with final $EC_{50}$ values of 25.1 nM and 46.1 nM, on par with the values for *Sf* RyR (Fig. 1b and Supplementary Table 1). This chimeric mutant RyR, named chiRyR, was utilized for further structural studies to reveal the interaction between insect RyR and diamide insecticides.

### Complex structure of chiRyR-FLU

We solved a cryo-EM structure of chiRyR in complex with FLU. chiRyR was recombinantly expressed in HEK293 cells and purified to homogeneity (Supplementary Fig. 2a, b). Since FLU is a channel activator, we reasoned that it may bind better to open channels and thus added a cocktail of RyR activators ($Ca^{2+}$/Caf/ATP/CaM1234) to help saturate its binding. CaM1234 also stabilizes the conformation of the RyR cytosolic cap, aiding with homogeneity and overall resolution.

The pore is in an open conformation, which is expected with a cocktail of activators including FLU (Fig. 3a, b). A masked refinement of the transmembrane region led to a local resolution of ~3.5 Å near the FLU binding site, allowing an unambiguous assignment of the FLU ligand (Supplementary Figs. 2d, e; Supplementary Fig. 3b; Supplementary Fig. 4b; Supplementary Table 3). The binding site is located at the interface between the transmembrane region and the large cytosolic cap of RyR (Fig. 2a). FLU is primarily stabilized by van der Waals interactions supplemented with hydrogen bonds (H-bonds) (Fig. 2b). It is coordinated by several residues from the transmembrane helices S1-S4, including Lys4563, Tyr4564, and Leu4567 from S1; Ile4657 from S2; Tyr4791, Ser4792, and Tyr4795 from S3; and Asp4815, Gly4819, and Val4820 from S4 (rRyR1 numbering) (Fig. 2b). Among them, Lys4563 and Asp4815 play key roles in sandwiching FLU by forming H-bonds from two opposite sides. Additionally, Arg4824 from the S4-S5 linker is also important in stabilizing the pose of the sulfide amine group via H-bonding.

### Complex structure of chiRyR-TET

To gain insights into the binding of both classes of diamide insecticides, we solved a cryo-EM structure of chiRyR in complex with TET in the presence of the same modulators ($Ca^{2+}$/Caf/ATP/CaM1234). After masking the transmembrane region, an unambiguous density of TET was visible, which also bound to the pVSD, with a local resolution of ~3.2 Å at the binding site (Fig. 2a; Supplementary Figs. 2d, e; Supplementary Fig. 3c; Supplementary Fig. 4c; Supplementary Table 3). This confirms that both classes of diamides generally occupy the same binding site, in contrast with previous reports suggesting that they bind to different regions[33–35].

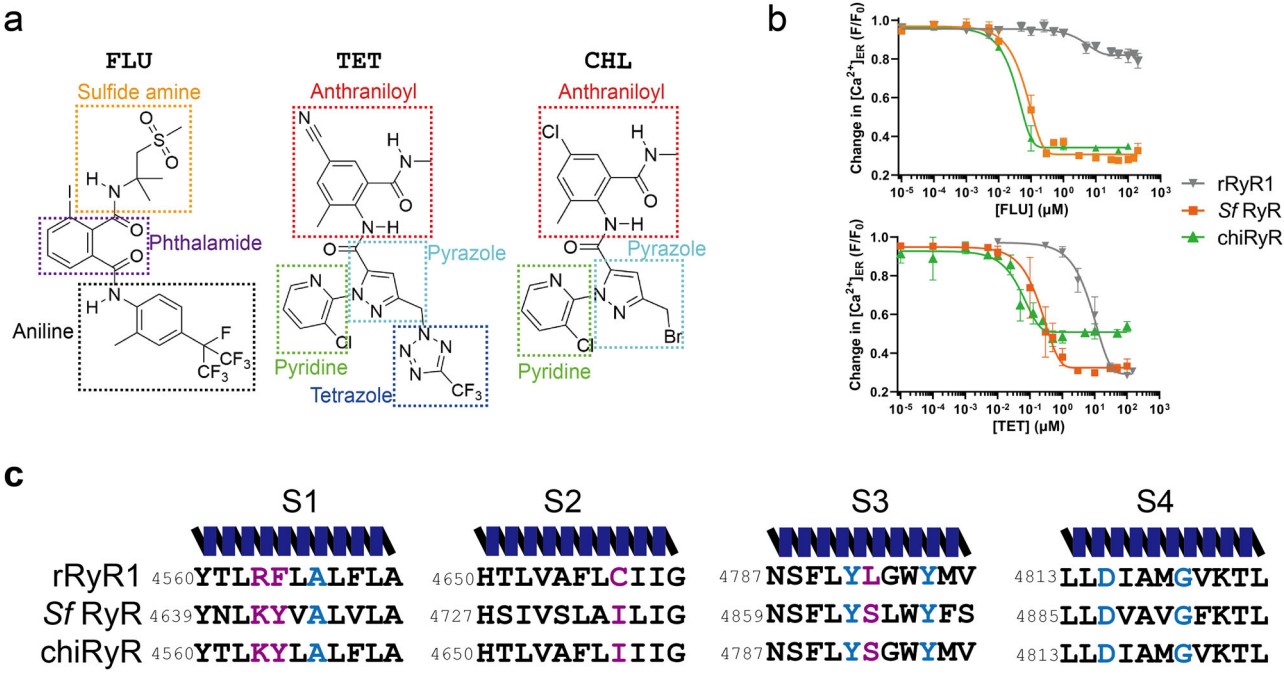

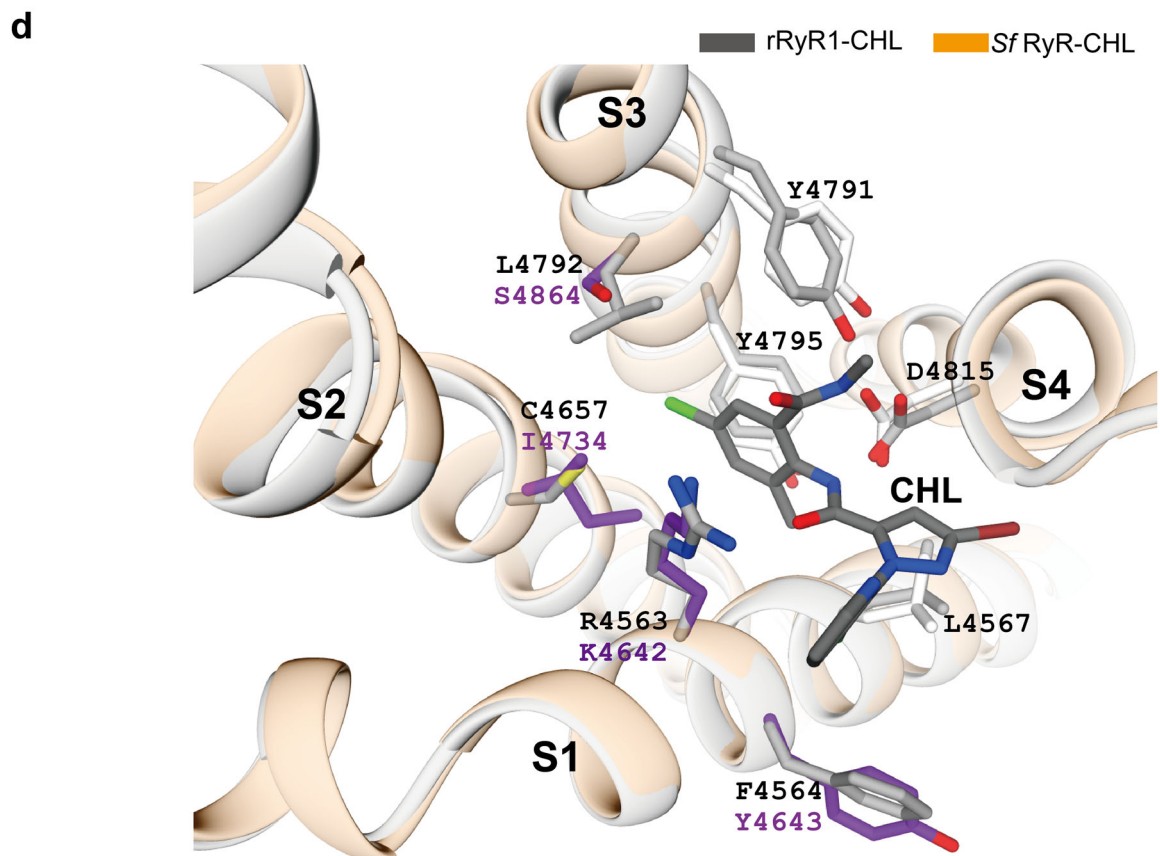

**Fig. 1 | Design of chiRyR. a** Chemical structures of FLU, TET, and CHL. **b** Dose-dependent effects of FLU (top) and TET (bottom) on $F/F_0$ in cell lines expressing rRyR1, *Sf* RyR, or chiRyR ($n = 3$). The average fluorescence intensity of the last 100 seconds (F) was normalized to that of the initial 100 seconds ($F_0$) to reflect the luminal $Ca^{2+}$ concentration change in the ER. The data are shown as the mean values ± SD. Source data are provided as a Source Data file. **c** Sequence alignment of diamide insecticide-binding sequences in RyRs. The four non-conserved residues involved in diamide coordination, which are selected to generate chiRyR, are colored in purple, and the other diamide-coordinating residues are colored in blue. **d** Comparison of the diamide-binding pocket in rRyR1 (white, PDB ID: 6M2W) and *Sf* RyR (orange, homology model based on 6M2W). The insect-specific diamide-coordinating residues are colored in purple.

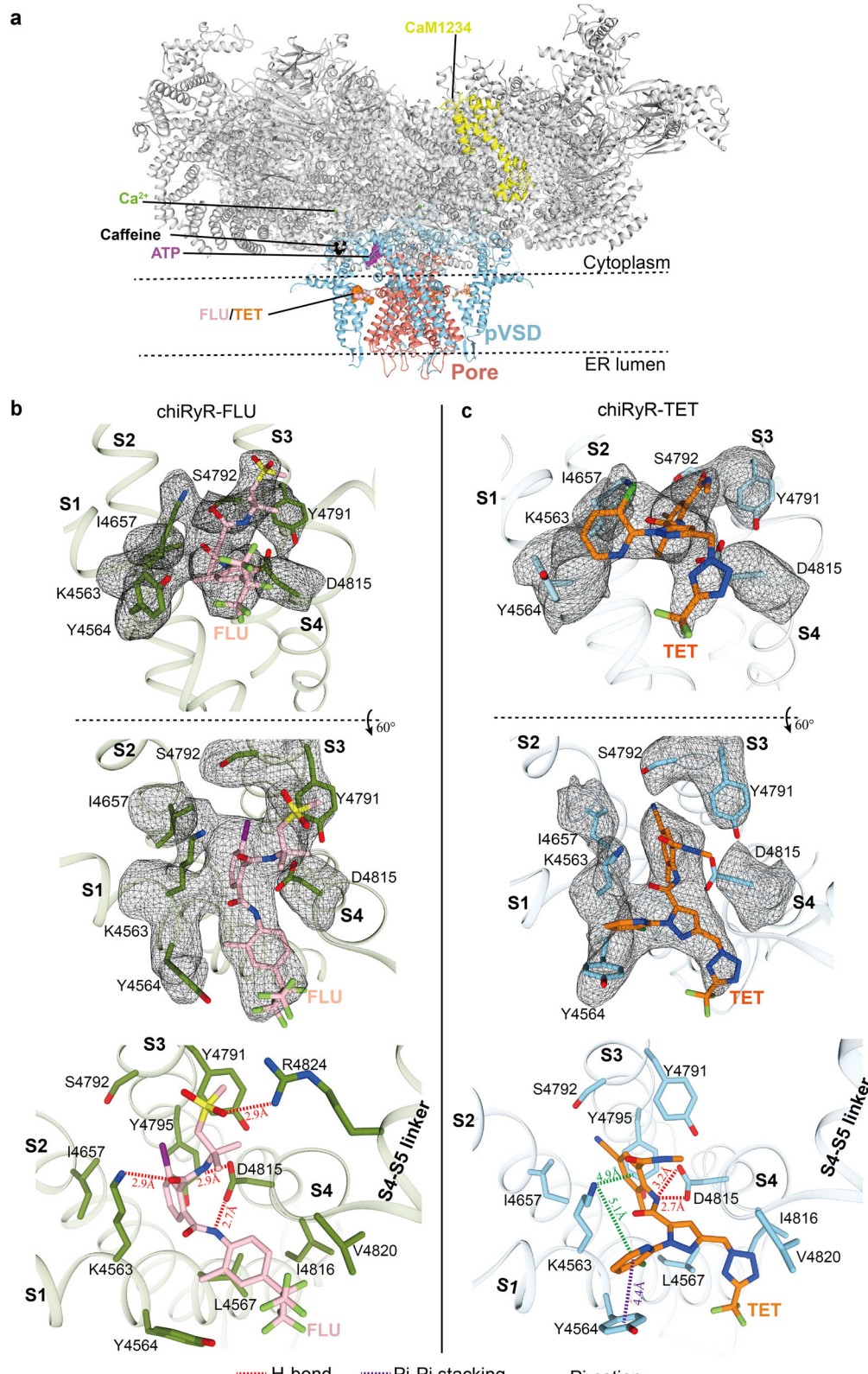

**Fig. 2 | Structures of chiRyR in complex with FLU and TET. a** Both FLU and TET bind to a site in the pVSD domain of the transmembrane region of chiRyR. **b, c** Enlarged views of the diamide-binding site from the chiRyR-FLU (**b**) and chiRyR-TET (**c**) complex structures. The densities of the ligands and coordinating residues are shown at the 5σ level in the up two panels, and the detailed interactions are displayed in the bottom panels. H-bonds, Pi-Pi stackings, and Pi-cation interactions are represented by red, purple, and green dashed lines, respectively.

However, there are distinct differences in how these two diamides engage the RyR (Figs. 2b, c; Supplementary Fig. 6a). First, Lys4563 and Asp4815 adopt conformations similar to those observed in the chiRyR-FLU complex but form different interactions with TET. Asp4815 forms two H-bonds with the nitrogen atom located in the amide that connects the anthraniloyl and pyrazole moieties of TET, in contrast to FLU, where the amides on both sides of the benzene ring participate in the hydrogen bonding. Second, the benzene ring of TET inserts deeper into the groove between the S2 and S3 helices, facilitating the formation of a cation–π interaction with Lys4563. Third, the pyridine ring of TET, which represents a major difference between the two families of diamide insecticides, forms several interactions that are absent in FLU, including a cation–π interaction with Lys4563 and π–π stacking with Tyr4564. Finally, the H-bond between Arg4824 from the S4-S5 linker and the sulfide amine group of FLU is absent in TET because of the substitution by a smaller amide group.

Within the anthranilic diamide insecticide family, there are two major structural differences between TET and CHL: 1) the substitution of the chlorine atom on the anthraniloyl moiety in CHL with a cyano group in TET and 2) the addition of a tetrazole moiety in TET, which replaces the bromine on the pyrazole moiety of CHL (Supplementary Fig. 6b). Both of these modifications might be associated with the broader insecticidal spectrum of TET. Among the four nonconserved residues between rRyR1 and $Sf$ RyR in the binding pocket, L4792S plays the most important role in selectivity toward insect RyR (Supplementary Table 2). Ser4792 forms direct contact with the cyano group of TET (Fig. 2c), which might contribute to species selectivity. Although the density of helix S0 is not well resolved in this structure, on the basis of superposition with the previous rRyR1 structure (PDB ID: 6M2W)[26], the bulky tetrazole moiety of TET should nestle into a space between helices S4 and S0, forming additional interactions with Leu4567 from S1 and Ile4816 and Val4820 from S4 (Fig. 2c). Future improvement of the local resolution in this region would aid in the accurate design of this moiety of ligands to further fine-tune the insecticidal spectrum of diamide insecticides.

## Diamide-induced RyR gating

To gauge the impact of diamide binding to chiRyR, we also obtained a structure in the absence of diamide (Supplementary Fig. 2d, e; Supplementary Fig. 3a; Supplementary Fig. 4a; Supplementary Table 3). This reference structure, termed ref-chiRyR, was solved under the exact same experimental conditions as the diamide-bound structures, including $Ca^{2+}$, caffeine, ATP, CaM1234, and 2% DMSO. In ref-chiRyR, Lys4563 and Asp4815 form a putative ionic interaction, which needs to be disrupted for FLU or TET binding. While Asp4815 maintains a similar conformation to the apo state, Lys4563 undergoes a significant conformational change upon diamide binding (Fig. 3c).

Notably, the structures of ref-chiRyR, chiRyR-FLU and chiRyR-TET all adopt an open conformation, as indicated by the distances between the sidechains of Ile4937 in diagonally opposed subunits (Figs. 3a, b). Since ref-chiRyR is in an open state, to understand the diamide-induced gating process, we also created a homology model of closed-state ref-chiRyR (ref-chiRyR-closed-HM) on the basis of the cryo-EM structure of closed-state rRyR1 solved in the presence of caffeine, ATP, and $Ca^{2+}$ (PDB ID 5TAQ). Compared with the ref-chiRyR, the binding of additional diamide ligands, FLU or TET, expands the pocket. This subsequently causes a displacement of the S4-S5 linker, which moves S5 outward to relax the constriction of the helical bundle in the pore (Figs. 3d–f). On the other hand, the binding of these ligands also induces an outward movement of helix S3. This pushes the S2-S3 linker connecting with the U-motif and indirectly transfers the conformational change to the C-terminal domain (CTD) of RyR, pulling open the channel by rotating the cytosolic end of the helix S6 (Figs. 3d–f). To accommodate the binding of the ligands, both the S4-S5 and S2-S3 linkers in chiRyR-FLU and chiRyR-TET undergo a larger displacement

than the one between ref-chiRyR-closed-HM and ref-chiRyR (Supplementary Movie 1). Furthermore, FLU induces a slightly greater displacement in the S4-S5 linker than TET does, probably due to the additional contact between Arg4824 from the linker and the sulfide amine moiety in FLU.

## Anthranilic diamide insecticides are more effective against resistant RyRs

Most of the reported resistance mutations against diamide insecticides cluster in a region near its binding site (Fig. 4a). Among them, I4790M and G4946E (DBM numbering) are the two most commonly observed resistance mutations and are often found together. When this double mutation was introduced into chiRyR (chiRyR-I4657M/G4819E), it increased the $EC_{50}$ for CHL by more than 5000-fold (from 1.6 nM to 14.6 μM) as measured by time-lapse experiments. This increase was comparable to that observed in $Sf$ RyR ($EC_{50}$ = 5.5 nM for $Sf$ RyR versus 9.1 μM for $Sf$ RyR- I4734M/G4891E) (Fig. 4b; Supplementary Table 4). Similar results were obtained via a [³H]ryanodine binding assay, which also revealed that the double mutations confer high resistance to CHL and TET (Fig. 4c; Supplementary Fig. 7b; Supplementary Table 5). In contrast, the double mutation almost completely abolished the activity of FLU according to the results of both assays (Figs. 4b, c; Supplementary Table 4; Supplementary Table 5), suggesting that these two families of diamide insecticides have very different effects and that the anthranilic acid diamide is more potent against the double-mutant resistant RyR (Fig. 8).

To test whether this observation extends to insects, we used a gene-edited *Drosophila melanogaster* carrying the M4758I mutation (rendering it equivalent to wild-type $Sf$ RyR). The $LD_{50}$ value for CHL for this transgene is 127.9 nM. Introducing the G4915E mutation to wild-type *Drosophila melanogaster* (the heterozygous mutation G4915E/G was introduced because the homozygous mutation is lethal), which makes it equivalent to the double-mutant $Sf$ RyR-I4734M/G4891E, increases this value by four orders of magnitude ($LD_{50}$ = 1.6 mM) (Fig. 4d; Supplementary Table 6). In contrast, FLU can no longer kill the same transgene but can retain a similar $LD_{50}$ for the transgene that only carries M4758I. Thus, these in vivo experiments qualitatively match the in vitro data.

## Structures of chiRyR harboring diamide resistance mutations

To determine whether these resistance mutations reduce binding affinity or cause structural changes that modulate channel gating, we recombinantly expressed and purified chiRyR-I4657M/G4819E (Supplementary Fig. 2a, c) and solved its cryo-EM structure at 3.6 Å resolution. The binding site was resolved at a local resolution of 3.2 Å (Supplementary Figs. 2d, e; Supplementary Fig. 3d; Supplementary Fig. 4d; Supplementary Table 3). The functional expression of chiRyR-I4657M/G4819E was confirmed by time-lapse experiments, which revealed similar $EC_{50}$ values for caffeine compared to chiRyR (Supplementary Fig. 7a, Supplementary Table 4). This finding was further confirmed by a $Ca^{2+}$-dependent [³H]ryanodine binding assay, which revealed biphasic $Ca^{2+}$-dependency similar to that of chiRyR (Supplementary Fig. 7b). These mutations significantly reduce the pocket volume. Additionally, G4819E introduces a net negative charge and alters the charge distribution of the pocket (Fig. 5c, e; Supplementary Fig. 8). This finding aligns with observations that G4946E causes stronger resistance compared to I4790M in DBMs and other lepidopteran pests[21,36].

To reveal the binding pose of diamides in RyRs with resistance mutations, we aimed to solve the cryo-EM structure of chiRyR-I4657M/G4819E in complex with a diamide insecticide. Because anthranilic acid diamide is more potent against this resistant RyR and CHL is more soluble than TET, we used a high concentration of CHL to saturate its binding to chiRyR-I4657M/G4819E and solved the corresponding cryo-EM structure. This allowed us to visualize the binding site at a local

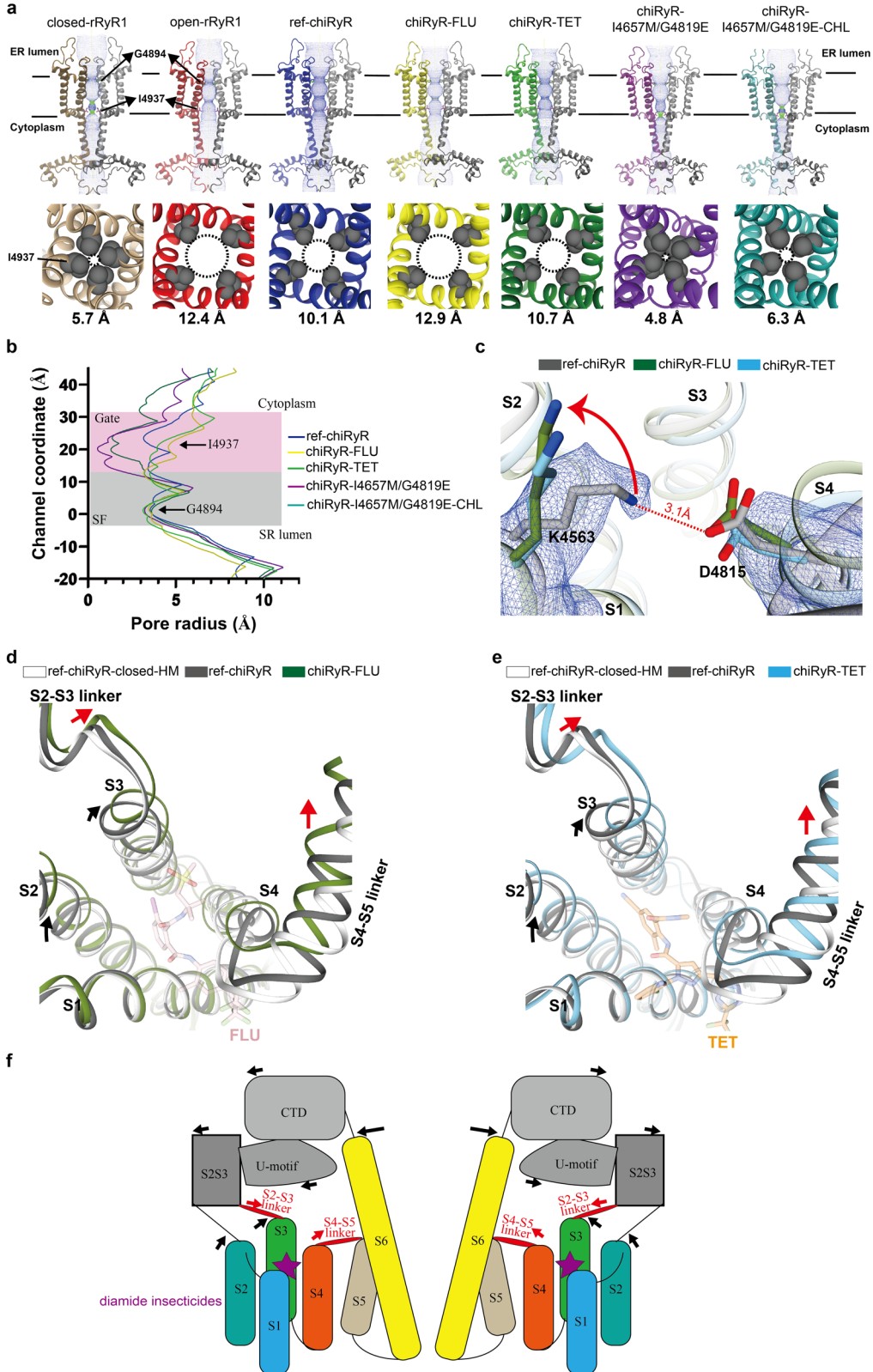

resolution of ~3.2 Å (Figs. 5a, b; Supplementary Figs. 2d, e; Supplementary Fig. 3e; Supplementary Fig. 4e; Supplementary Table 3).

Compared with the unbound structure of chiRyR-I4657M/G4819E, the binding of CHL did not significantly change the conformation of Met4657 but induced an ~90-degree rotation of the side chain of Glu4819 toward the cytosol to avoid the steric hindrance (Fig. 5c, Supplementary Movie 2). Both mutations form

direct van der Waals contacts with CHL: Met4657 contacts the benzene ring and the chlorine atom from the anthraniloyl moiety, whereas Glu4819 contacts the distal amide group (Fig. 5b). These contacts result in a shift of the anthraniloyl moiety toward the groove between the S3 and S4 helices while keeping the pyridine and pyrazole groups relatively unchanged compared with a homology model of chiRyR in complex with CHL (chiRyR-CHL-HM; based

**Fig. 3 | FLU/TET-induced RyR opening. a** Side view (top) and top view (bottom) of the pore forming domains of rRyR1 and chiRyRs. Structures compared include closed-state rRyR1 (PDB ID: 5TAQ), open-state rRyR1 (PDB ID: 5TAL), ref-chiRyR, chiRyR-FLU, chiRyR-TET, chiRyR-I4657M/G4819E, and chiRyR-I4657M/G4819E-CHL. The ion permeation pathway through the transmembrane pore is denoted by green dots (areas accessible to single $H_2O$) and blue dots (areas accessible to double $H_2O$). The diameters shown at bottom are the distances between two closest atoms from Ile4937 at diagonal positions of the channel gate. **b** Graph showing pore radii of different rRyR1 and chiRyRs structures, with the pore radius plotted against the channel coordinate. Selectivity filter (SF) and channel gate regions are highlighted by gray and purple shading, respectively. **c** Salt-bridge formed between Lys4563 and Asp4815 is broken upon the binding of FLU or TET. The densities of Lys4563 and Asp4815 in ref-chiRyR are shown at the 5σ level. The movement of Lys4563 is indicated by a red arrow, while the interaction between Lys4563 and Asp4815 is shown with a red dashed line. **d, e** Comparison of the diamide-binding sites in closed ref-chiRyR (ref-chiRyR-closed-HM: a homology model based on closed rRyR1 (PDB 5TAQ)), open ref-chiRyR, and chiRyR-FLU (**d**) chiRyR-TET (**e**). The binding of FLU/TET induces displacements in pVSD, S2-S3 and S4-S5 linkers compared to both closed and open ref-chiRyRs. The superposition was performed based on helices S1. The movements of transmembrane helices are illustrated by arrows. **f** Schematic cartoon showing that the binding of diamide insecticides induces the opening of the channel by pulling S2-S3 and S4-S5 linkers.

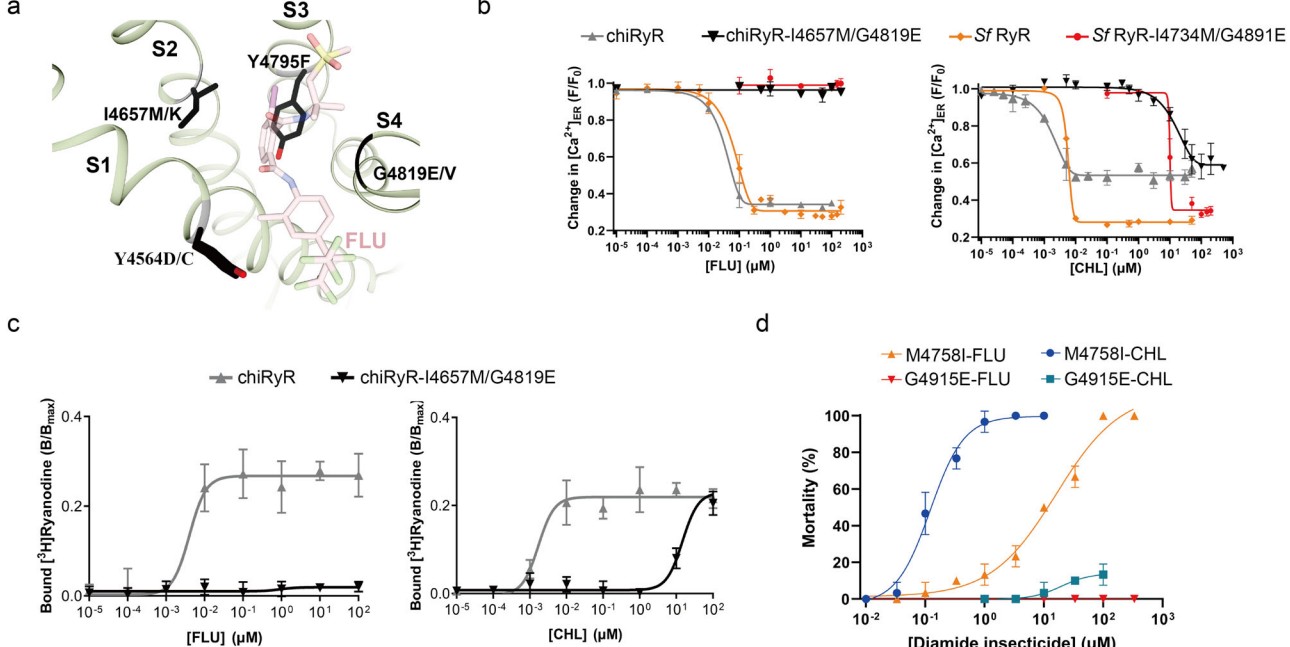

**Fig. 4 | Mutations in RyR cause resistance to diamide insecticides. a** Locations of the resistance mutations in RyR. Mutation sites, colored in black, cluster near the diamide binding site. **b, c** Double resistance mutations reduce the sensitivity of chiRyR to FLU (left) and CHL (right) as shown by time-lapse $[Ca^{2+}]_{ER}$ measurements (**b**) and [³H]ryanodine binding assays (**c**). The data are shown as the mean values ± SD (n = 3 for time-lapse; n = 4 for [³H]ryanodine binding). Source data are provided as a Source Data file. **d** FLU/CHL toxicity curves for gene-edited *Drosophila* (n = 30) expressing RyR M4758I (corresponding to *Sf* RyR WT) or RyR G4915E (corresponding to *Sf* RyR-I4734M/G4891E). Mortality rates were calculated after 24 h of feeding with either FLU or CHL. The data are shown as the mean values ± SD. Source data are provided as a Source Data file. While the resistance mutations reduce the potency of both insecticides, CHL is relatively more potent compared to FLU against the resistant *Drosophila*.

on the rRyR1-CHL cryo-EM structure as a template; PDB ID: 6M2W) (Fig. 5d).

The superposition of the structures of chiRyR-FLU and chiRyR-I4657M/G4819E-CHL revealed that the G4819E mutation would form a major clash with the sulfide amine moiety of FLU. This explains why the phthalic acid diamide insecticides are not effective against resistant insects (Supplementary Fig. 6c).

**Diamide-resistant mutations result in a closed-state structure**

The structures of chiRyR-I4657M/G4819E, regardless of the presence of CHL, all adopt a closed pore (Figs. 3a, b). This is unexpected because they were obtained in the presence of known RyR activators, including $Ca^{2+}$, caffeine, ATP, and CaM1234, which typically result in a large proportion of open channels in the case of rRyR1[37]. This finding contrasts directly with the fact that mainly open channels were obtained for the chiRyR under the same conditions. Thus, the exact identity of the residues in the pVSD, which contains the binding pocket, has a significant effect on channel gating. However, our functional experiments do not show a decreased sensitivity for caffeine or activating $Ca^{2+}$ (Supplementary Fig. 7). Although a

previous report has suggested that the structure of RyR1 is very similar in detergent versus in a lipid environment[38], the presence of particular lipids or additional protein binding partners in the functional assays may be responsible for this discrepancy. Further experiments will be needed to find the exact cause but, if true, a stabilized closed state would create an additional energetic barrier for the diamide insecticides to overcome to trigger channel opening. Notably, to the best of our knowledge, this structure represents the first instance of a diamide insecticide bound to a closed RyR, albeit for a resistant mutant. Verification is needed to determine whether this binding and its impact on channel gating are also possible for wild-type insect RyRs.

## Discussion

Parallel to antibiotic resistance in microorganisms, insecticide resistance in pests has become a global problem. The continued use of diamide insecticides has created intense evolutionary pressure, leading pests in different parts of the world to develop very similar resistance mutations in their RyR, a telltale example of convergent evolution.

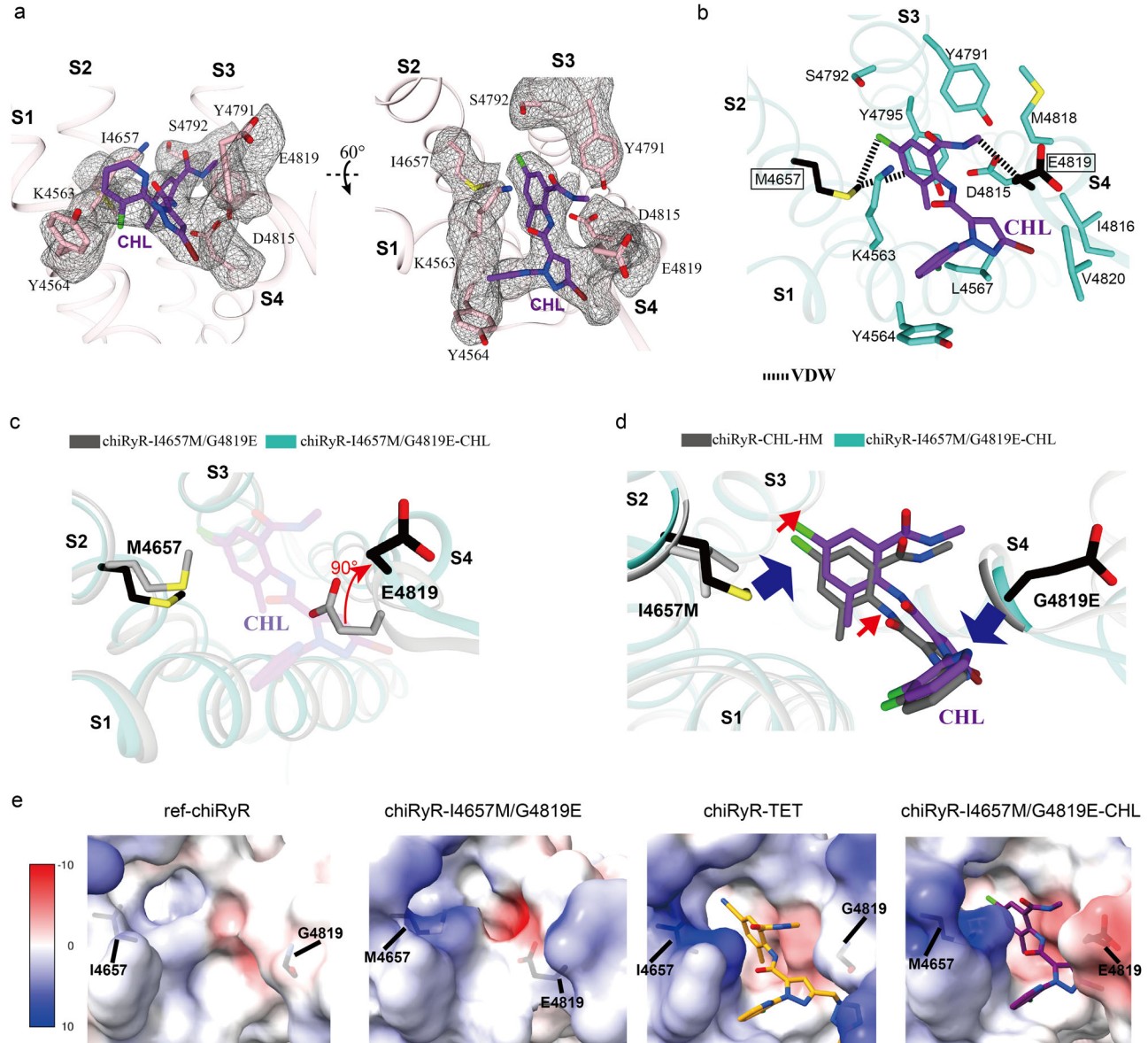

**Fig. 5 | Structures of resistant chiRyRs. a, b** Enlarged views of the diamide-binding site from the chiRyR-I4657M/G4819E-CHL complex structure. The densities of the ligands and coordinating residues are shown at the 5σ level in panel (**a**), and the detailed interactions are displayed in panel (**b**). **c** Comparison of the diamide-binding site between chiRyR-I4657M/G4819E and chiRyR-I4657M/G4819E-CHL. The side chain of Glu4819 rotates ~90 degrees upon CHL binding, as indicated by the red arrow. **d** Comparison of the diamide-binding site between chiRyR-CHL (chiRyR-

CHL-HM: a homology model based on rRyR1-CHL (PDB ID: 6M2W)) and chiRyR-I4657M/G4819E-CHL. The resistance mutations cause a displacement of the ligand, indicated by the red arrows. **e** The shape and charge distribution of the diamide binding sites from ref-chiRyR, chiRyR-I4657M/G4819E, chiRyR-TET, and chiRyR-I4657M/G4819E-CHL are compared. The positively and negatively charged areas are colored in blue and red, respectively. The resistance mutations make the pocket smaller and more negatively charged.

To study the resistance mechanisms, we first introduced four pest-specific mutations into the diamide-binding site of rabbit RyR. This reproduces the high-affinity binding to diamides, with the L4792S mutation making the largest contribution. Since obtaining large amounts of purified insect RyRs has been cumbersome, this approach provides an opportunity to study the specific binding mode of diamide insecticides via cryo-EM. This finding showed that the insecticides adopt a distinct conformation in the chiRyR complex structure compared with our previous studies in which wild-type rRyR1 was used[26].

It has been controversial whether the two major classes of diamide insecticides, phthalic acid and anthranilic diamide insecticides, bind to different sites on RyR and modulate the channel in a species-specific manner[34,39]. Previous studies using radiolabeled ligands have indicated that these two classes of diamide insecticides may bind to

distinct but coupled sites on RyR[35]. Our complex structures of rRyR1-CHL, chiRyR-FLU, and chiRyR-TET reveal that three diamide insecticides from both classes indeed target the same binding site in the pVSD of RyR. However, while two anthranilic diamide insecticides share a similar binding pose, the phthalic acid diamide FLU adopts a distinct one. The difference in binding modes contributes significantly to the difference in insecticidal spectrum and efficacy against resistant pests.

Our studies revealed that the diamide insecticides induce distinct structural changes upon binding. Pushing residues in the binding pocket results in a widening of the pVSD, which leads to movement of the S2-S3 and S4-S5 linker helices and subsequent pore opening. Although the details differ, these observations are parallel to voltage-sensing mechanisms in voltage-gated ion channels, where movements

of the VSDs are often linked to pore opening through the S4-S5 linker. Interactions between Arg4501 from S1, Tyr4720 from S3, and Asp4744 from S4 of mouse RyR2 (corresponding to Arg4563, Tyr4791, and Asp4815 in rRyR1) are critical for stabilizing the closed state of the channel, and their substitution by alanine strongly activates RyR2[40]. Our structures show that the binding of diamide insecticides disrupts these important interactions, opening the channel, which aligns with previous observations.

A major goal in the development of the next-generation diamide insecticides is to improve their safety and protect nontarget species. Our mutagenesis results suggest that among the four nonconserved residues in the diamide-binding pocket, the L4792S mutation contributes the most to the selectivity toward insect RyRs. Ser4792 forms direct contacts with the cyano group of the anthraniloyl moiety of TET and the iodine group of FLU. Thus, future modifications targeting these moieties could lead to further improvements in species-selectivity.

Unsurprisingly, pests have evolved resistance through direct mutations in the diamide binding site. The next generation of insecticides need therefore address the structural changes imparted by the mutations. To gain deeper insights and aid in the future design of such compounds, we investigated the two most prevalent resistance mutations in agricultural pests: I4790M and G4946E. Our structure revealed that these mutations alter the local structure and polarity of the diamide-binding pocket, leading to the reduction in binding affinity. Based on our complex structure, which shows that CHL binds to the resistance mutant pocket in a different conformation but also disrupts the S1-S3-S4 interaction, future designs should focus on achieving shape and charge complementarity with the resistance mutant pockets to enhance affinity over CHL.

However, unexpectedly, these mutations also stabilize the channel in a closed state, rendering activation of the RyR by the diamide agonizts more difficult. This stabilizing effect was only observable through cryo-EM structures when RyR was extracted using detergents. In the membrane environment, however, caffeine sensitivity and $Ca^{2+}$-dependent [³H]ryanodine binding properties were similar between the wild-type and resistant mutant channels. This suggests that the double mutant does not inherently stabilize the closed state under native conditions but does reduce the potency of diamides. While DMSO was required for the cryo-EM experiments, we cannot entirely rule out the possibility that DMSO had an additional effect on the particle distribution. To the best of our knowledge, this is the first report of insecticide resistance involving the dynamic equilibrium of an ion channel. This finding imposes a significant challenge for the potency of next-generation compounds. It also raises the question about the impact of this reduced activation ability on resistant insects in the absence of diamide insecticides. For instance, the homozygous G4946E mutation has been found to be lethal in some insect species, such as *Drosophila melanogaster*[22], likely due to excessive loss of function of RyR. However, in other species, such as *Spodoptera exigua* and *Plutella xylostella*, the same mutation is tolerated, suggesting that specific residues elsewhere in the RyR can compensate. The diamide-binding pocket is crucial for channel activity, yet many insects manage to survive without any significant fitness cost despite resistance mutations that alter the pocket while maintaining the S1-S3-S4 interaction essential for function. This could result from natural selection acting on a large number of natural mutations, reflecting a high degree of adaptability in insects. Overall, there appears to be a delicate balance between mutations that reduce overall fitness but confer stronger resistance. For example, the recently identified I4790K in RyR shows a higher resistance than G4946E with a lower fitness cost, based on net replacement rates[23,41,42].

Although the potencies can be improved through modification of the diamides directly targeting the resistant pockets, alternative strategies could also be considered. With a molecular weight exceeding 2 MDa, the insect RyR represents an abundance of potential binding sites for molecules that could further activate them. Although diamide insecticides trigger channel opening, a potent channel blocker could also be considered. For example, ryanodine and the scorpion toxin imperacalcin can cause full or partial blockage of pores[37,43–47]. Finally, small molecules can interfere with the binding of important binding partners that are known to regulate RyRs[48–50]. However, the main challenge with all of these scenarios consists of finding novel pockets that are unique to insects and preferably unique to pests. Thus, the structures of full-length insect RyRs and their complexes would provide immense value.

In summary, our cryo-EM structures of wild-type and resistant chiRyR in complex with diamide insecticides provide insights into the selectivity of different types of diamide insecticides and the mechanism of resistance mutations. Our work may enable rational structure-based design of the next generation of green insecticides with the desired high potency, selectivity, and antiresistance properties.

## Methods

### Generation of stable inducible HEK293 cell lines
Flp-In T-REx HEK293 cell lines stably expressing R-CEPIA1er, an engineered ER-targeting fluorescent $Ca^{2+}$-sensing protein, were generated via the Jump-In system[29,51]. Constructs, including Wild-type rabbit RyR1 (rRyR1), rRyR1 containing one or more insect-mimicking mutations (R4563K, F4564Y, C4653I, and F4792S, rRyR1 numbering), chiRyR containing two resistance mutations (I4657M and G4819E, rRyR1 numbering), wild-type *Spodoptera frugiperda* RyR (*Sf* RyR), and *Sf* RyR containing two resistance mutations (I4734M and G4891E, *Sf*RyR numbering), were cloned and inserted into the pcDNA5/FRT/TO vector. RyR mutants were generated via mutagenesis using a QuikChange mutagenesis kit (Stratagene) following the manufacturer's instructions, with primer sequences provided in Supplementary Table 7, and the mutated fragments were used to replace the corresponding fragments in the wild-type constructs. Flp-In T-REx HEK293 cells stably expressing R-CEPIA1er were co-transfected with the cloned pcDNA5/FRT/TO and pOG44 at a 1:1 ratio using Lipofectamine 3000 (Invitrogen). The cells were transferred to new dishes on the second day with a fresh medium. After cell attachment, hygromycin was added at a final concentration of 100 µg/mL. The medium supplemented with antibiotics was changed every four days until single clones were identified. Positive clones were verified by western blot and calcium imaging assays.

### Western blotting
Western blotting was performed as described[52]. Briefly, HEK293 cells stably expressing chiRyR or chiRyR-I4657M/G4819E were plated on 6-well cell culture plates, and protein expression was induced with 2 µg/mL doxycycline for 24 h. Then, the cells were harvested and rinsed twice with PBS. Proteins were extracted with Pro-Prep protein extraction solution (iNtRON Biotechnology), separated on 3–12% linear gradient polyacrylamide gels, and transferred to PVDF membranes. The membrane was probed with primary antibodies against RyR1 (F-1, Santa Cruz Biotechnology, at a dilution of 1:1000) and calnexin (C4731, Sigma-Aldrich, at a dilution of 1:5000), followed by HRP-conjugated anti-mouse IgG (04-18-18, KPL, at a dilution of 1:5000) and anti-rabbit IgG (074-1516, KPL, at a dilution of 1:5000), respectively. Positive bands were detected by chemiluminescence using ImmunoStar LD (Fujifilm Wako Chemicals) as a substrate.

### Time-lapse [$Ca^{2+}$]$_{ER}$ measurement
A total of $2 \times 10^4$ HEK293 cells stably expressing R-CEPIA1er and RyRs were seeded into a single well of a 96-well black, clear-bottom plate (Coring). The inducer doxycycline was added at a final concentration of 2 µg/mL, 12 h after seeding. After 24 h, the medium was replaced with a HEPES-buffered Krebs solution (5 mM HEPES pH 7.4, 140 mM

NaCl, 5 mM KCl, 2 mM $CaCl_2$, 1 mM $MgCl_2$, and 11 mM glucose). Fluorescence signal measurements were performed using a FlexStation 3 fluorometer (Molecular Devices) according to a published method[29]. The R-CEPIA1er was excited at 560 nm and emitted at 610 nm. Time-lapse images were captured every 10 seconds for 300 seconds. Caffeine (0.1-100 mM) was dissolved in a HEPES-buffered Krebs solution, and $10^{-5}$–$10^2$ μM diamide insecticides were dissolved in the same buffer with 1% DMSO. All the compounds were added to the cells 100 seconds after the start of recording. The average fluorescence for the first 100 seconds ($F_0$) and the last 100 seconds ($F$) was used to calculate the fluorescence change induced by the compounds ($F/F_0$). Each measurement was repeated three times.

## [³H]Ryanodine binding
[³H]Ryanodine binding was performed as previously described with some modifications[40,52]. The isolated HEK293 microsomes were incubated at 25 °C with 5 nM [³H]ryanodine in a medium containing 0.17 M NaCl, 20 mM 3-(N-morpholino)−2-hydroxypropanesulfonic acid (MOPSO) at pH 7.0, 5 mM AMP, 2 mM dithiothreitol and various concentrations of free $Ca^{2+}$ buffered with 10 mM ethylene glycol-bis(2-aminoethylether)-N,N,N′,N′-tetraacetic acid (EGTA). WEBMAXC STANDARD[53] was used for calculation of free $Ca^{2+}$ concentrations. After 5 h incubation, samples were filtered through polyethyleneimine-treated GF/B filters using a Micro 96 Cell Harvester (Skatron Instruments) to obtain the protein-bound [³H]ryanodine binding ($B$). Nonspecific binding was determined by the addition of 20 μM unlabeled ryanodine. Separately, the maximum number of [³H]ryanodine binding ($B_{max}$) was determined via Scatchard plot analysis using various concentrations (3–20 nM) of [³H]ryanodine in a high-salt medium containing 1 M NaCl. The resulting $B/B_{max}$ represents the average activity of each mutant.

## Expression and purification of FKBP12.6 and CaM1234
Human FKBP12.6 was cloned and inserted into a pET-His6-GST-TEV vector (Addgene, plasmid #29707). Human CaM1234, a mutant calmodulin with four mutations (D20A, D56A, D93A, and D129A) that prevent calcium binding[54], was cloned and inserted into a modified pET28 vector with an N-terminal 6xHis-tag, an MBP-fusion protein, and a TEV cleavage site. Both plasmids were transformed into BL21 (DE3) cells and cultured in 2YT culture media at 37 °C. Protein expression was induced with 0.4 mM isopropyl β-D-thiogalactoside (IPTG) when the $OD_{600}$ reached ~0.6, and the cells were cultured at 30 °C for 6 h. The cells were harvested by centrifugation at $5000 \times g$ for 15 minutes at room temperature. For FKBP12.6, harvested cells were sonicated in buffer A (10 mM HEPES, pH 7.4, 250 mM KCl) with 25 μg/mL DnaseI, 25 μg/mL lysosome, 10 mM 2-mercaptoethanol (βME), 1 mM PMSF, and 20 mM imidazole. The lysate was centrifuged at $12,000 \times g$ for 30 minutes, and the supernatant was filtered through a 0.22 μm filter, then loaded onto a gravity Ni-NTA column (High-affinity Ni-charged resin FF, GenScript) and washed with 10-column volumes of buffer A. Target proteins were eluted with buffer A containing 500 mM imidazole, dialyzed against buffer B (10 mM Tris-HCl, pH 8.8) for 3 h, loaded onto an anion exchange column (HiPrep Q HP, GE Healthcare Life Sciences), and eluted with a linear gradient of 0–50% of buffer B with 1 M KCl. Fractions containing FKBP12.6 were concentrated and loaded onto a gel-filtration column (Superdex 200 pg, GE Healthcare Life Sciences) and eluted with buffer A. CaM1234 protein was purified similarly to FKBP12.6. After applying the lysate to a gravity Ni-NTA column, the protein was cleaved overnight by TEV at 4 °C. His-MBP-CaM1234 was applied to an amylose column to separate His-MBP tags from CaM1234. The CaM1234 protein was dialyzed for 3 h in buffer A with 10 mM EDTA, loaded onto a phenyl-Sepharose column, and eluted with buffer A containing 10 mM $CaCl_2$. The fractions containing CaM1234 were injected into a gel filtration column (Superdex 75 pg, GE

Healthcare Life Sciences) and eluted with buffer A. Both proteins were concentrated to approximately 5 mg/mL and stored at −80 °C.

## Purification of chiRyR and chiRyR-I4657M/G4819E
chiRyR (rRyR1-R4563K + F4564Y + C4653I + F4792S) or chiRyR-I4657M/G4819E were integrated into HEK293 cells with an inducible promoter and grown in DMEM supplemented with fetal bovine serum (FBS). At approximately 90% confluency, the cells were induced with 2 μg/mL doxycycline (Sigma) for three days. For each purification experiement, a total of 150 dishes (150 mm petri dishes) of cells were used. Cells were suspended in a lysis buffer (20 mM Na-MOPS pH 7.4, 500 mM NaCl, 2.5% CHAPS, 1.25% PC, 2 mM DTT, and 1× protease inhibitor cocktail) and lysed by sonication at 75% amplitude for a total of 12 minutes, with 1 second on and 2 seconds off. After 1 h of incubation at 4 °C, 5 mg His-GST-FKBP12.6 was added to the solution. Following another 1 h incubation, the mixture was ultracentrifuged (Hitachi CP100 ultracentrifuge with a rotor of S/N 2520) at $150,000 \times g$ for 1 h. The supernatant was filtered through a 0.45 μm filter, mixed with 3 mL of GS4B resin (GE Healthcare Lifesciences) equilibrated with buffer C (20 mM Na-MOPS, pH 7.4, 500 mM NaCl, 0.5% chaps, 2 mM DTT, and 1× protease inhibitor cocktail), and stirred at 4 °C for 3 h. The mixture was poured into a gravity column and washed with 30 mL of buffer C. Approximately 4 mL of buffer C containing 3 mg of TEV protease was added to the column for overnight on-column cleavage. For chiRyR, the eluents were concentrated to 500 μL, loaded onto a gel filtration column (Superose 6 10/300 GL, GE Healthcare Life Sciences), and eluted with a buffer containing 20 mM Na-MOPS, pH 7.4, 500 mM NaCl, 0.025% Tween-20, 30 μM $CaCl_2$, 2 mM DTT, and 1 x protease inhibitor cocktail. For chiRyR-I4657M/G4819E, an anion exchange purification step replaced the gel filtration purification step to reduce protein loss during concentration. After the FKBP12.6 affinity purification step, the eluents were diluted to a 100 mM NaCl concentration and loaded onto Capto HiRes Q 5/50 column. The target protein was eluted at a 600 mM NaCl with 0.025% Tween-20. The target proteins were verified by 8% SDS-PAGE, and the peak fractions containing tetrameric RyRs were collected, concentrated to ~5 mg/mL, and stored at −80 °C.

## Cryo-EM sample preparation
For all datasets, 5 mg/ml chiRyR samples were mixed with an activating cocktail to reach final concentrations of 5 mM caffeine, 2 mM ATP and 100 μM CaM1234. Diamide insecticides (dissolved in DMSO) or pure DMSO were then added to achieve a consistent final concentration of 2% DMSO across all the wild-type chiRyR samples, ensuring uniform conditions. For the wild-type chiRyR, either FLU or TET was added at final concentrations of 50 μM. For the resistant chiRyR-I4657M/G4819E, a higher final concentration 625 μM CHL required 5% DMSO, but the sample without CHL still contained only 2% DMSO. The samples were incubated for 1 h at room temperature. 3 μL of each sample was applied to holey carbon grids with 2 nm carbon (QUANTIFOIL R1.2/1.3, Au 300). Grids were blotted for 5–8 s with a blot force of 3 applied and subsequently flash-frozen with liquid ethane using a Vitrobot Mark IV (Thermo Fisher Scientific)[37].

## Cryo-EM data acquisition
All five datasets were collected on a 300 kV FEI Titan Krios instrument equipped with a Gatan K3 direct electron detector, using the automated collection system EPU (Thermo Fisher Scientific) at ×22,500 magnification and 1.06 Å per pixel. Micrographs were recorded in superresolution mode with a total electron dose of 50 e⁻/ Å² and 32 frames for each stack. The defocus values were in the range of −1 to −3 μm. The total numbers of movies collected were: 3826 (ref-chiRyR), 4365 (chiRyR-FLU), 5718 (chiRyR-TET), 5937 (chiRyR-I4657M/G4819E), and 5290 (chiRyR-I4657M/G4819E-CHL) (Supplementary Fig. 4).

## Image processing

Diagrams for the data processing are presented in Supplementary Fig. 4. Image processing was performed using CyroSPARC v3.3[55]. All movies were imported, followed by full-frame motion correction and contrast transfer function (CTF) estimation using CTFFIND4.1 within CryoSPARC. Due to ice contamination or drift, 291, 88, 37, 25, and 25 movies were eliminated from the ref-chiRyR, chiRyR-FLU, chiRyR-TET, chiRyR-I4657M/G4819E, and chiRyR-I4657M/G4819E-CHL datasets, respectively. The map of rRyR1 with CHL (EMD30067) was imported into CryoSPARC and used to create a template for particle picking. Results from template picking were subjected to 2 cycles of 2D classification. The selected particles in the five datasets were 54,414 (ref-chiRyR), 69,627 (chiRyR-FLU), 99,274 (chiRyR-TET), 34,441 (chiRyR-I4657M/G4819E), and 97,482 particles (chiRyR-I4657M/G4819E-CHL). These particles were used for the 3D reconstruction. To filter low-quality particles, 2 cycles of initial reconstruction and heterogeneous refinement were applied, except for the chiRyR-I4657M/G4819E dataset, where one cycle yielded better results. For the ref-chiRyR dataset, a 3D classification step was applied to effectively separate the open and closed states. However, due to the low number of particles, the closed state could not be refined to generate a high-resolution map. A similar step did not generate clear distinct states for other datasets and thus was not included. Good classes were used for homogeneous refinement with C1 symmetry and non-uniform refinement with C4 symmetry. Local CTF refinement was applied, followed by another non-uniform refinement (C4 symmetry). Reference-based motion correction was then applied, followed by another non-uniform refinement to improve global resolution. To enhance the local resolution of the ligand binding, a mask was generated in the transmembrane domain using Chimera[56]. The mask was used for local refinement in CryoSPARC[55]. The maps were further sharpened in PHENIX using autoSHARP[57].

## Model building and structure refinement

The complex structure of rRyR1 with CHL in the presence of $Ca^{2+}$/ATP/caffeine/CaM1234 (PDB ID: 6M2W) was used as a template for building the models of all five structures. The template was fit to the maps of five datasets. The models were adjusted to fit into the maps, the ligand model was built, and the mutations in the pocket were performed in Coot[58]. Structure refinement was performed using PHENIX real-space refinement[57]. Additionally, the ISOLDE modular in chimeraX was used to resolve clashes[59]. The refinement and validation statistics are shown in Supplementary Table 3.

## Pore radius calculations

The radius of each pore was calculated using HOLE[60]. The pore regions (residues 4820-5036) from the five structures, including ref-chiRyR, chiRyR-FLU, chiRyR-TET, chiRyR-I4657M/G4819E, and chiRyR-I4657M/G4819E-CHL, were analyzed. Five structures were superposed using UCSF Chimera, and the pore dimension calculations were performed in HOLE with default parameters. Images of the pore radius were generated using VMD software[61].

## Bioassay of Drosophila melanogaster

Mutant *Drosophila* strains (M4758I, G4916E) were generated using CRISPR-Cas9 gene editing technology[25]. For each insecticide concentration, 30 flies were collected and transferred into three 5-ml glass vials (ten flies per vial). The flies were fed through capillaries containing 10% (w/v) sucrose water mixed with the diamide insecticides. Mortality was assessed 24 h after feeding. Data was statistically analyzed, and the $LC_{50}$ was calculated and analyzed with GraphPad Prism 8.0 (GraphPad, San Diego, CA, USA).

## Reporting summary

Further information on research design is available in the Nature Portfolio Reporting Summary linked to this article.

## Data availability

Cryo-EM maps and structures generated in this study have been deposited in the EMDB and PDB databases, respectively, with the following accession codes: ref-chiRyR: PDB 8XLF, EMDB EMD-38447 and EMD-60900 (TMD local). chiRyR-FLU: PDB 8XJI, EMDB EMD-38398 and EMD-38551 (TMD local). chiRyR-TET: PDB 8XKH, EMDB EMD-38417 and EMD-38553 (TMD local). chiRyR-I4657M/G4819E: PDB 8XLH, EMDB EMD-38448 and EMD-60899 (TMD local). chiRyR-I4657M/G4819E-CHL: PDB 8Y40, EMDB EMD-38908 and EMD-60901 (TMD local). The structures used in this paper are available in the PDB database under accession codes 6M2W, 5TAL, and 5TAQ. All the data needed to evaluate the conclusions are presented in the paper or the Supplementary Materials. Source data are provided in this paper.

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

## Acknowledgements

This research was funded by the National Key Research and Development Program of China (no. 2022YFE0108400 to Z.Y.), the National Natural Science Foundation of China (no. 32372580 to Z.Y. and no. 32272576 to S.W.), the Haihe Laboratory of Sustainable Chemical Transformations (no. 24HHWCSS00005 to Z.Y.), the JSPS KAKENHI (no. 23K24067 to T.M.) and the Naito Foundation (to. T.M.).

## Author contributions

Conceptualization: L.L. and Z.Y.; Methodology: L.L., C.W., T.M., F.V.P., and Z.Y.; Investigations: L.L., C.W., W.W., H.J., T.M. and T.K.; Resources: Z.Y.; Data analysis: L.L., Y.S.C.; Writing—original draft: L.L. and Z.Y.; Writing—review and editing: L.L., H.H., Y.W., S.W., T.M., F.V.P. and Z.Y.; Supervision: Z.Y.; Project administration: Z.Y.; Funding acquisition: S.W., T.M., and Z.Y.

## Competing interests

The authors declare no competing interests.

## Additional information

[1]Tianjin Key Laboratory for Modern Drug Delivery and High-Efficiency, Frontiers Science Center for Synthetic Biology, School of Pharmaceutical Science and Technology, Faculty of Medicine, Tianjin University, Tianjin, China. [2]Haihe Laboratory of Sustainable Chemical Transformations, Tianjin, China. [3]Institute of Plant and Food Science, Department of Biology, Southern University of Science and Technology, Shenzhen, Guangdong, China. [4]Department of Cellular and Molecular Pharmacology, Juntendo University Graduate School of Medicine, Tokyo, Japan. [5]Department of Biochemistry and Molecular Biology, Life Sciences Institute, University of British Columbia, Vancouver, British Columbia, Canada. [6]College of Plant Protection, State & Local Joint Engineering Research Center of Green Pesticide Invention and Application, Nanjing Agricultural University, Nanjing, Jiangsu, China. [7]Syngenta Jealott's Hill International Research Centre, Bracknell, Berkshire, UK. [8]Guangdong Laboratory for Lingnan Modern Agriculture (Shenzhen Branch), Agricultural Genomics Institute at Shenzhen, Chinese Academy of Agricultural Sciences, Shenzhen, Guangdong, China. [9]These authors contributed equally: Lianyun Lin, Changshi Wang. ✉e-mail: filip.vanpetegem@gmail.com; yuchi@tju.edu.cn

