## [Transparent Peer Review file · Nature Communications]

Cryo-EM structures of ryanodine receptors and diamide insecticides reveal the mechanisms of selectivity and resistance

Corresponding Author: Professor Zhiguang Yuchi

Version 1:

Reviewer comments:

Reviewer #1

(Remarks to the Author)

In the manuscript titled "Cryo-EM Structures of Ryanodine Receptors and Diamide Insecticides Revealing Mechanisms for Selectivity and Resistance" by Lin et al., the authors delve into the impact of different diamide insecticides on their molecular target, RyR channels, and explore the mechanisms through which RyR mutations impart diamide resistance. Previously, the authors identified a diamide binding site in mammalian RyR channels, although mammalian receptors typically do not bind the insecticide unless exposed to high concentrations.

Due to the non-identity of residues within the binding pocket of insect RyRs and the challenges associated with expressing insect RyR, the authors opted to engineer an 'insect-like' diamide binding pocket on the background of the rabbit RyR1 sequence. Furthermore, additional chimeric RyR channels were generated to incorporate mutations known to confer diamide resistance. Functional analysis of these chimeric RyRs indicated Ca²⁺ store depletion and a diamide affinity comparable to that of insect RyR, making them suitable candidates for further structural investigations via cryoEM.

While these studies offer valuable insights into the structural implications of insecticides on RyR function and the mechanisms through which mutations can confer insecticide resistance - crucial for the rational design of future insecticides - there are specific points outlined below that need to be addressed prior to publication of the manuscript.

(1) A comprehensive sequence alignment comparing rabbit RyR and sfRyR would provide valuable insights into sequence variations beyond the pVSD/diamide binding site. This alignment could shed light on how these differences might influence the proposed mechanism in insect cells.

(2) Line 148-149: I'm uncertain about the rationale behind why this channel activator may bind more readily to the open state of the channel. Is this already established? Do you have cryoEM structures of the chiRyR with only the diamide drug and not other activators? Additionally, I'm curious if the diamide drugs bind to the insect RyR channel in the closed state.

(3) Given that the TM domains are resolved at 3.8 Å, it's likely that this resolution would allow for the visualization of side chains along the ion conduction pathway. In light of this, why are the pore dimensions presented as distances between C-alphas? What are the pore radii when calculated with HOLE?

(4) It is hard to ascertain if all residues interpreted in the diamide binding pocket are supported by cryoEM densities. Therefore, it would be essential to show cryoEM densities for both the protein and bound ligand within the diamide binding pocket for each of the presented maps.

(5) Although the authors normalize the background fluorescence in their Ca²⁺ measurements among the different cell lines, it remains unclear whether the data were normalized for RyR protein expression levels. To ensure the validity of their findings, it's crucial to conduct Western blots for each cell line. These blots are necessary and critical for determining if the observed effects stem from varying expression levels between the cell lines. It's worth noting that low expression levels may be a concern for the chiRyR-C4657M/G4819E clone, as indicated by the small peak on the SEC.

(6) It's intriguing that the chiRyR-C4657M/G4819E structure appears closed in the presence of activators. Do you have any direct evidence indicating that the channels are functional? It's plausible that the rabbit RyR channel may have reached its limit regarding the number of mutations tolerated in this region. Alternatively, is it possible that you were unable to discern the class of open channels from this dataset?

(7) Have the RyR mutant channels ever been tested for 3H-ryanodine binding? Conducting a direct functional assay of chiRyR-C4657M/G4819E would provide clarity regarding the functionality of the protein and help determine the parameters required to open the channel. Furthermore, since this mutant was generated based on the majority of the rabbit RyR sequence, it appears overly speculative to assert that the mutations introduce an additional barrier to activation without additional functional data.

(8) Figure 5 lacks clarity regarding the information it intends to convey. Is it meant to serve as a summary or a model? Additionally, the orientation for reading the figure (left/right or up/down) isn't clearly defined. It should be easy for the reader to discern which structures were resolved and which are part of the proposed model. Regarding the state of chiRyR, is it open or closed? Was the closed state of the chiRyR channel resolved? Furthermore, listing ▲ C4657M/G4819E as a ligand seems questionable — is there a specific rationale behind this? All structures were solved in the presence of caffeine, why is the caffeine not depicted in several structures shown in Figure 5, e.g. the chiRyR-C4657M/G4819E structure?

Minor:

- In Figure S1, a structure is labeled as chiRyR-C4657M/G4819E-apo. The term "apo" conventionally denotes the absence of exogenous ligands. However, Table S1 indicates that all structures were generated in the presence of activating ligands. This discrepancy should be addressed to ensure clarity.
- The manuscript lacks consistency in terminology and naming conventions concerning the various structures and the bound ligands/drugs. This inconsistency makes it particularly challenging to interpret Figure 5.
- Line 131: Please provide clarification regarding the activator(s) used in the functional experiments.
- Lines 133-134, the EC50 values mentioned in the text are not consistent with those in Table 1.
- Line 139, the first 3 individual residues did not exhibit a significant decrease in EC50 value for FLU, as per Table 2. It appears that rRyR1 F4564Y has the most substantial effect on CHL, while L4792S has the greatest impact on FLU. However, this observation seems to be omitted from any discussion regarding the influence of these residues on ligand binding. Additionally, Table 1 does not summarize the data for the combined mutant.
- Lines 200 and 203: 2d instead of 2e?
- Lines 246-247: 'Most of the reported resistance mutations against diamide insecticides cluster in a region near its binding site.' Incorporating a figure that maps the mutations to the model around the binding site would be beneficial.
- Line 249 states that "the two most commonly observed resistance mutations" are C4657M/G4819E. However, in the introduction (line 105), it's mentioned that "I4790M and G4946E are the most common mutations found...", though in lines 278-279, it's noted that the residues are equivalent. This equivalence should be clarified at the beginning to avoid confusion and potential misinterpretation.
- Line 256: there is no Supplementary Figure 7
- Line 264: the values reported in the text and the table data do not align
- Line 282 and line 291: the cited figures do not correspond to the appropriate figure panels.
- Line 326 and 328: The residues listed as LD50 or LC50 in Table 7 and the text are not consistent.
- The legend for Figure 2 does not match the panel labelings
- Figure S1: to be consistent with the figure legend, in panel a, FKBP12.6 needs to be shown or labeled in the right lane of the gel
- Line 214: what are 'important differences'?
- Line 370: should use 'among' instead of 'out of'?
- Figure 2a, it would be easier to compare the chemical structures if CHL was flipped 180 degrees to make the color box in the same order as TET
- Consistency in color coding for maps and models across all figures would enhance convenience and ease of understanding.
- In Figure 4, the label for E4819 should be more accurate; the last one appears blue, not red. For panel f, the purple arrow should be made more visible. In panel g, labels should be added for the two models.

Reviewer #2

(Remarks to the Author)

Cryo-EM structures of Ryanodine Receptors and diamide insecticides reveal the mechanism for selectivity and resistance

The authors provide an analysis of the structure and function of chiRyR, a modified version of the rabbit ryanodine receptor type 1 protein, in the presence of two diamide insecticides, flubendiamide (FLU) and tetraniliprole (TET), highlighting the same binding site of FLU and TET in the pVSD domain of the transmembrane region of chiRyR. They also discuss the functional and structural impact of diamide resistance mutations in chiRyR, which decrease the effectiveness of FLU and CHL. The authors compare the diamide-binding pocket between the wild-type chiRyR and the chiRyR-C4657M/G4819E mutant and provide a schematic representation of chiRyR modulation by FLU, CHL, and the resistant mutations. Overall, the

manuscript provides valuable insights into the molecular mechanisms underlying the binding and modulation of chiRyR by diamide insecticides, contributing to the understanding of pesticide resistance in insects and the development of more effective insecticides. However, there are major technical and analytical issues that need to be fixed and addressed.

- 1) The authors need to explain why recombinant expression of insect RyRs is challenging. They should also show the homology between insect RyR and the mammalian RyRs.
- 2) Figure 2a should be figure 1a. and the structure of CHL should be flipped vertically so it's easier to compare to TET.
- 3) In general, the figure panels should be better organized and ordered to correspond to the text.
- 4) The claimed local resolution of 3.2Å in figure S2 is not convincing since the scale goes from 3.2Å to 7.2Å. The subpanel should be focused on the ligand binding site and the scale should be from 3Å to 4Å to show that local resolution is approximately 3.2Å and not 3.82Å and 3.71Å as it is shown in figure S1e. Methodologically, the authors could improve the local resolution by expanding the particles on a C4 basis, creating a mask of one of the binding sites (one protomer only), and performing local refinement with the expanded particles and the local mask.
- 5) Figure S3 should be part of the main manuscript since it is fundamental to understand the good fit of the atomic model, which it is not supported. The fit of the ligands with a resolution of 3.2-3.8Å is poor and figure S3 is not convincing. Looking at the cryo-EM maps, TET and FLU are well placed, but the conformation of the ligand "flexible" motives cannot be precisely determined. In the case of CHL, the fit is worse than the other two, suggesting that although it is clear that CHL is there, the pose of the ligand cannot be precisely determined. This needs to be address and any conclusions at the atomic level should be avoided or made with caution.
- 6) The MM-GBSA analysis is based on an atomic model with low confidence. MM-GBSA analysis is useful when the atomic model is accurate. At a resolution of 3.2-3.8Å, neither the ligands or the chiRyR sidechains can be confidently modelled. Any conclusions coming from this analysis should be made with caution. Some results are contradictory like in the case of chiRyR-C4657M/G4819E-CHL. My recommendation is to remove the whole MM-GBSA analysis from the manuscript since it adds no real value.
- 7) It is claimed that Lys4563 and Asp 4815 form an ionic interaction. They should show the distance between sidechains and the cryo-EM maps to show that the sidechains are confidently placed. Unconvincing at a 3.72Å resolution.
- 8) Ref-chiRyR processing is incomplete. As shown in figure 3a, the pore is in an intermediate conformation, suggesting a mixture of open and closed channels, as highlighted in line 230. This is confirmed by the relative low intensity of the pore in the cryo-EM map, suggesting a mix of conformations. Therefore, all comparisons to ref-chiRyR are incorrect. It is inappropriate to use this structure as a reference. The authors need to perform local 3D variability or 3D classification with TMD mask in order to separate the closed from the open particles. Once separated the authors need to build a model for the closed state and a model for the open state. The authors should also perform local 3D variability or 3D classification with TMD mask of the other conditions to convince the readers that there are no mixed closed/open conformations.
- 9) The comparison in Figure 3b-f is poorly done. They compare the closed state of rRyR1 to the open state of chi-RyR1 in the presence of FLU and TET. There are too many variables changing. The changes in the helices positions are mostly due to the change from the closed to the open state. The authors should compare the open state of chi-RyR1 in the presence of FLU and TET to the open state of ref-chiRyR. The authors should also show how these helices change in wt-rRyR1 or ref-chiRyR going from the closed to the open state. Only then can the real impact of the ligands be assessed. It might be that the ligands stabilize the open state, without inducing conformational changes.
- 10) In figure 4b, the mutant is called chiRyR-C4657M/G4819E but the residue shown is I4657. Since, the residue in chiRyR is I4657, because it was already mutated from rRyR1, the mutant should be named chiRyR-I4657M/G4819E.
- 11) In figure 4b, the distance measurement between poorly defined sidechains due to the resolution of 3.6Å is inappropriate. It is better to show the cryo-EM maps and show experimentally how the pocket looks smaller. It would be interesting to know how this pocket changes between the closed and open states.
- 12) In figure 4e, the authors claim a movement of the Glu4819 sidechain. The authors should show the cryo-EM maps to support this claim, otherwise is not convincing due to the resolution.
- 13) In figure 4f, the authors claim a movement of CHL. However, looking at the cryo-EM map, the CHL ligand does not fit the cryo-EM density well enough. Due to the poor fit, any conclusions made regarding the movement of such ligand are unconvincing.
- 14) The text mentions figure S7 which does not exist.
- 15) Figure 2f and 4f are hard to understand.
- 16) In all figures where a distance or interaction is shown, a label with the value of such distance should be included.
- 17) The caffeine concentration used in the time-lapse experiments is not mentioned. A representative curve of the experiment should be shown.
- 18) In line 483, it says "HEK293 cells stably expressing chiRyR" but in line 485 it says "cells were induced". Are the cells stably expressing or inducible?
- 19) The purification method is different for chiRyR1 and the mutant. Why is that? No explanation is given.
- 20) 5% of DMSO is used for the mutant chiRyR in presence of CHL. The authors claimed it is needed to reach 625uM CHL, however the experiment in figure 4a shows that 100uM is enough for inhibition, suggesting 100uM can be used and 2% DMSO can also be used. DMSO promotes the closing of the channel, so any conclusions regarding the state of the channel in 5% DMSO should be taken with caution and clarified in the manuscript.
- 21) What is the logic of using a base model (PDB 6M2W) for the chiRyR and a completely different model with lower resolution and no CaM1234 (PDB 5TAQ) for the mutant chiRyR? It is not hard to build the model of the closed state from the open state, and the comparison would be more consistent and scientifically correct.

(Remarks to the Author)

This is a very well written manuscript elucidating critical aspects of Diamide insecticide binding in their molecular targets and the precise role of insecticide resistance mutations, by using advanced Cryo-EM methodology. The study is comprehensive and it will guide future studies in the field of insecticide resistance, including expansions in vectors of diseases (such as malaria vectors). The detailed analysis of the binding site among different active ingredients of the same (major) diamide insecticide class, provides extremely useful information and evidence for insect pest control applications and resistance management.

Some implications in medical applications relevant with RyRs are also nicely discussed in the manuscript.

A possible limitation one might consider, the insect-like chimeric RyRs (containing diamide binding pocket from insect and the remaining part from rabbit). However, as clearly explained, this is not a real issue, as all relevant mechanistic aspects are relevant. The missing structure of a diamide insecticide bound to a closed RyR for wild type (but not resistant mutant) would be good to be added, if logistics for addition possible and realistic (timeframe and feasibility).

Finally, the session (line 298) Diamide-resistant mutations stabilise the closed state of RyR could be better explained, so that generalist can easily digest this important information (not very clear).

Version 2:

Reviewer comments:

Reviewer #1

(Remarks to the Author)

The authors have adequately addressed the points raised in the initial review, resulting in significant improvements to the manuscript. The figures have been substantially enhanced, offering clearer and more informative visual representation of the data. The authors have also refined the manuscript's content, providing a more cohesive narrative that effectively communicates the study's findings and significance. These improvements make the manuscript well-suited for a broad scientific audience, enhancing its accessibility and impact.

Reviewer #2

(Remarks to the Author)

the revised manuscript addresses the concerns of the reviewers and is significantly improved.

Dear Reviewers,

We thank you for the constructive comments and suggestions, which have led to an overall improvement in our efforts to communicate the results. I have reproduced your comments verbatim below together with our responses. For clarity, your criticisms and/or suggestions are coloured in dark red, while our response is shown in dark blue.

Reviewer #1 (Remarks to the Author):

In the manuscript titled "Cryo-EM Structures of Ryanodine Receptors and Diamide Insecticides Revealing Mechanisms for Selectivity and Resistance" by Lin et al., the authors delve into the impact of different diamide insecticides on their molecular target, RyR channels, and explore the mechanisms through which RyR mutations impart diamide resistance. Previously, the authors identified a diamide binding site in mammalian RyR channels, although mammalian receptors typically do not bind the insecticide unless exposed to high concentrations.

Due to the non-identity of residues within the binding pocket of insect RyRs and the challenges associated with expressing insect RyR, the authors opted to engineer an 'insect-like' diamide binding pocket on the background of the rabbit RyR1 sequence. Furthermore, additional chimeric RyR channels were generated to incorporate mutations known to confer diamide resistance. Functional analysis of these chimeric RyRs indicated Ca²⁺ store depletion and a diamide affinity comparable to that of insect RyR, making them suitable candidates for further structural investigations via cryoEM.

While these studies offer valuable insights into the structural implications of insecticides on RyR function and the mechanisms through which mutations can confer insecticide resistance - crucial for the rational design of future insecticides - there are specific points outlined below that need to be addressed prior to publication of the manuscript.

COMMENT #1: A comprehensive sequence alignment comparing rabbit RyR and sfRyR would provide valuable insights into sequence variations beyond the pVSD/diamide binding site. This alignment could shed light on how these differences might influence the proposed mechanism in insect cells.

RESPONSE #1: We thank the reviewer for the suggestion. We have added a sequence alignment between rabbit RyR1 and Sf RyR, including the sequence identities of each domain, in Supplementary Fig. 1. We also included the identity of their pVSD domain in the introduction (lines 114-117): "However, diamide insecticides exhibit ~100-1,000 times greater selectivity for insect RyRs than for mammalian RyRs, and the binding site, located in the pseudo-voltage sensing domain (pVSD), is not conserved between insects and mammals, with approximately 49.5% sequence identity (Supplementary Fig. 1)." Additionally, we provided the overall identity and TM identity in the results (lines 130-132): "The overall sequence identity between Sf RyR and rRyR1 is 45.2%, with higher identity (~60%) in their transmembrane regions, where the diamide binding site is located (Supplementary Fig. 1)."

COMMENT #2: Line 148-149: I'm uncertain about the rationale behind why this channel activator may bind more readily to the open state of the channel. Is this already established? Do you have cryoEM structures of the chiRyR with only the diamide drug and not other activators? Additionally, I'm curious if the diamide drugs bind to the insect RyR channel in the closed state.

RESPONSE #2: We thank the reviewer for the insightful questions. The rationale for using known activators to stabilize the open state of RyR and increase the occupancy of other activators have been well-documented in several studies. For example, Haji-Ghassemi et al. demonstrated that introducing more activating conditions resulted in a higher fraction of rabbit RyR1 channels bound to IpCa, an RyR-targeting scorpion toxin (Sci Adv 9, eadf4936, 2023). This provides a foundation for our approach.

We currently do not have a cryo-EM structure of chiRyR in complex with only the diamide insecticide and no other activators. However, insights from our previous studies are relevant (Nat Chem Biol 16, 1, 2020). In our earlier work, we solved the structure of rabbit RyR1 in complex with chlorantraniliprole (CHL) both in the presence and absence of other activators (CaM1234, caffeine, ATP). In both scenarios, CHL stabilized the channel in an open-state conformation, and no closed-state structures were obtained during classification. The complex structure in the presence of other activators exhibited higher resolution, likely due to increased particle homogeneity. This is why, in the current study, we determined the structure in the presence of other activators to ensure optimal resolution and data quality.

Given the higher potency of diamide insecticides against insect RyR and chiRyR, we hypothesize that these compounds would stabilize the channels even more effectively than in mammalian RyR. This likely results in a predominant open-state conformation, preventing the observation of closed-state structures even in the absence of other activators.

The diamide binding site changes conformation upon channel gating and diamide binding. Our data suggest that diamide insecticides preferentially bind to the open-state pocket, which prevents the structural characterization of the complex in the closed-state. However, binding assays using fluorescent CHL and lipid bilayer experiments from other groups have demonstrated the diamide insecticides can bind to closed-state RyRs with a lower affinity compared to open-state RyRs, and increase the P_o of the channel (Biochemical and Biophysical Research Communications 508, 633–639, 2019). In summary, while diamide insecticides are capable of binding the closed-state RyRs, their affinity is significantly lower, making it difficult to capture the closed-state structure using cryo-EM.

COMMENT #3: Given that the TM domains are resolved at 3.8 Å, it's likely that this resolution would allow for the visualization of side chains along the ion conduction pathway. In light of this, why are the pore dimensions presented as distances between C-alphas? What are the pore radii when calculated with HOLE?

RESPONSE #3: After reprocessing the datasets using the new reference-based motion correction feature in cryoSPARC v4.4 and performing local refinements using a TMD mask, the resolutions of the transmembrane regions have been further improved to as high as 3.28 Å. We have modified the Figure 3 according to the reviewer's suggestion, presenting the pore dimensions using the distance between the side chains of Ile4937 at the gate. The pore diameters/radii calculated with HOLE are now shown in Figure 3a,b.

COMMENT #4: It is hard to ascertain if all residues interpreted in the diamide binding pocket are supported by cryoEM densities. Therefore, it would be essential to show cryoEM densities for both the protein and bound ligand within the diamide binding pocket for each of the presented maps.

RESPONSE #4: We have added new panels in Figure 2b,c and Figure 5a to display the cryo-EM densities for both the ligands and the coordinating residues in the diamide binding pockets of chiRyR-FLU, chiRyR-TET, and chiRyR-I4657M/G4819E-CHL. Additionally, we now also show the density for the S1-S6 helices in Supplementary Figure 5. This should help visualize and support the modeled protein residues within the binding pockets.

COMMENT #5: Although the authors normalize the background fluorescence in their Ca^{2+} measurements among the different cell lines, it remains unclear whether the data were normalized for RyR protein expression levels. To ensure the validity of their findings, it's crucial to conduct Western blots for each cell line. These blots are necessary and critical for determining if the observed effects stem from varying expression levels between the cell lines. It's worth noting that low expression levels may be a concern for the chiRyR-C4657M/G4819E clone, as indicated by the small peak on the SEC.

RESPONSE #5: To compare the expression levels of chiRyR and chiRyR-I4657M/G4819E, we followed the reviewer's suggestion and performed both Western blots (WB) and [3H]Ryanodine binding experiments. The WB results show that the expression level of chiRyR is only ~30% lower than that of chiRyR-I4657M/G4819E (Supplementary Figure 2a), which is supported by the B_{max} values obtained from the [3H]Ryanodine binding assay (Supplementary Figure 7b). We normalized the $[Ca^{2+}]_{ER}$ measurement fluorescence signals for RyR expression levels as the reviewer suggested. Despite the expression level would affect the scale of fluorescence signals, in principle it shouldn't impact any EC50 value. The purification protocols for chiRyR and chiRyR-I4657M/G4819E were slightly different: after the affinity purification step, an ion exchange column (HiResQ) was used to purify chiRyR-I4657M/G4819E, while a SEC column (Superose6) was used to purify chiRyR. Although the elution profiles look different, the peak areas of the two proteins are similar, suggesting similar purification yields and expression.

COMMENT #6: It's intriguing that the chiRyR-C4657M/G4819E structure appears closed in the presence of activators. Do you have any direct evidence indicating that the channels are functional? It's plausible that the rabbit RyR channel may have reached its limit regarding the number of mutations tolerated in this region. Alternatively, is it possible that you were unable to discern the class of open channels from this dataset?

RESPONSE #6: To demonstrate chiRyR-I4657M/G4819E is functional, we performed the following experiments:

1. [³H]Ryanodine binding assay, which is routinely used and correlates with channel open probability.
 - We showed that chiRyR-I4657M/G4819E can be activated by low, and inhibited by high Ca²⁺, similar to chiRyR (Supplementary Figure 7b).
 - We also showed that chiRyR-I4657M/G4819E can be activated by CHL and TET, with EC₅₀ values of 14.7 μM and 87.8 μM, respectively (Supplementary Table 5, Figure 4c).
2. Time-lapse experiments.
 - We showed that chiRyR-I4657M/G4819E can be activated by caffeine with an EC₅₀ value of 7.1 mM, which is similar to the EC₅₀ values of chiRyR (19.5 mM) (Supplementary Table 4, Supplementary Figure 7a), and rRyR1 (23.7mM, our previous data that is not shown in this paper).
 - We also showed that chiRyR-I4657M/G4819E can be activated by CHL with EC₅₀ values of 14.6 μM (Supplementary Table 5, Figure 4b).

We included these findings in the manuscript:

- Lines 256–262: “When this double mutation was introduced into chiRyR (chiRyR-I4657M/G4819E), it increased the EC₅₀ for CHL by more than 5,000-fold (from 0.4 nM to 14.6 μM) as measured by time-lapse experiments. This increase was comparable to that observed in Sf RyR (EC₅₀ = 5.5 nM for Sf RyR versus 9.1 μM for Sf RyR- I4734M/G4891E) (Fig. 4b; Supplementary Table 4). Similar results were obtained via a [³H]Ryanodine binding assay, which also revealed that the double mutations confer high resistance to CHL and TET (Fig. 4c; Supplementary Fig. 7b, c; Supplementary Table 5).”
- Lines 285–289: “The functional expression of chiRyR-I4657M/G4819E was confirmed by time-lapse experiments, which revealed similar EC₅₀ values for caffeine compared to chiRyR (Supplementary Fig. 7a, Supplementary Table 4). This finding was further confirmed by a Ca²⁺-dependent [³H]Ryanodine binding assay, which revealed biphasic Ca²⁺-dependency similar to that of chiRyR (Supplementary Fig. 7b).”

After extensive data processing, including local 3D variability analysis and 3D classification with and without transmembrane masking, we could not obtain any open-state structure for chiRyR-I4657M/G4819E. We believe this discrepancy between the functional and structural results is likely due to the differences in the membrane environment. In [³H]ryanodine binding and the time-lapse experiments, RyRs are in a membrane environment, while for cryo-EM studies, RyRs are in detergents. The resistance mutations might destabilize the open state of RyR, but the membrane environment protects the channel from closing. In the presence of detergents, this destabilizing effect is amplified.

We added this to the discussion of the paper:

- Lines 397–403: “This stabilizing effect was only observable through cryo-EM structures when RyR was extracted using detergents. In the membrane environment, however, caffeine sensitivity and Ca²⁺-dependent [³H]ryanodine binding properties were similar between the wild-type and resistant mutant channels. This suggests that the double mutant does not inherently stabilize the closed state under native conditions but does reduce the potency of diamides.”

COMMENT #7: Have the RyR mutant channels ever been tested for 3H-ryanodine binding? Conducting a direct functional assay of chiRyR-C4657M/G4819E would provide clarity regarding the functionality of the protein and help determine the parameters required to open the channel. Furthermore, since this mutant was generated based on the majority of the rabbit RyR sequence, it appears overly speculative to assert that the mutations introduce an additional barrier to activation without additional functional data.

RESPONSE #7: As mentioned above, we performed [³H]ryanodine binding assay, following the reviewer’s suggestion, and demonstrated the functional effect of the mutations alongside the structural evidence. Additionally, we acknowledged the need for additional verification in wild-type insect RyRs by including a statement in results (lines 336-337) “Verification is needed to determine whether this binding and its impact on channel gating are also possible for wild-type insect RyRs.”

COMMENT #8: Figure 5 lacks clarity regarding the information it intends to convey. Is it meant to serve as a summary or a model? Additionally, the orientation for reading the figure (left/right or up/down) isn't clearly defined. It should be easy for the reader to discern which structures were resolved and which are part of the proposed model. Regarding the state of chiRyR, is it open or closed? Was the closed state of the chiRyR channel resolved? Furthermore, listing

▲C4657M/G4819E as a ligand seems questionable — is there a specific rationale behind this? All structures were solved in the presence of caffeine, why is the caffeine not depicted in several structures shown in Figure 5, e.g. the chiRyR-C4657M/G4819E structure?

RESPONSE #8: We thank the reviewer for the suggestion. We acknowledge that this figure is somehow misleading and have removed this figure to avoid any confusion it might cause.

Minor:

COMMENT #1: In Figure S1, a structure is labeled as chiRyR-C4657M/G4819E-apo. The term "apo" conventionally denotes the absence of exogenous ligands. However, Table S1 indicates that all structures were generated in the presence of activating ligands. This discrepancy should be addressed to ensure clarity.

RESPONSE #1: We have changed the label to chiRyR-I4657M/G4819E to resolve this discrepancy and ensure clarity.

COMMENT #2: The manuscript lacks consistency in terminology and naming conventions concerning the various structures and the bound ligands/drugs. This inconsistency makes it particularly challenging to interpret Figure 5.

RESPONSE #2: We thank the reviewer for pointing this out. We have standardized the naming of the five structures to: ref-chiRyR, chiRyR-FLU, chiRyR-TET, chiRyR-I4657M/G4819E, and chiRyR-I4657M/G4819E-CHL.

COMMENT #3: Line 131: Please provide clarification regarding the activator(s) used in the functional experiments.

RESPONSE #3: The activators mentioned in the functional experiments should be "diamide insecticides". We have updated the results section to reflect this change.

COMMENT #4: Lines 133-134, the EC50 values mentioned in the text are not consistent with those in Table 1.

RESPONSE #4: The values in Supplementary Table 1 are the correct ones. We have corrected the EC50 values in the text to match those in Table 1.

COMMENT #5: Line 139, the first 3 individual residues did not exhibit a significant decrease in EC50 value for FLU, as per Table 2. It appears that rRyR1 F4564Y has the most substantial effect on CHL, while L4792S has the greatest impact on FLU. However, this observation seems to be omitted from any discussion regarding the influence of these residues on ligand binding. Additionally, Table 1 does not summarize the data for the combined mutant.

RESPONSE #5: The impacts of these key residues responsible for species selectivity have been discussed extensively in our previous paper published in Nature Chemical Biology (2020). The results for the combined mutant are shown in the rows labeled "chiRyR" in Supplementary Table 1.

COMMENT #6: Lines 200 and 203: 2d instead of 2e?

RESPONSE #6: We have corrected this in the revised version.

COMMENT #7: Lines 246-247: 'Most of the reported resistance mutations against diamide insecticides cluster in a region near its binding site.' Incorporating a figure that maps the mutations to the model around the binding site would be beneficial.

RESPONSE #7: We have added a new panel (Figure 4a) in the revised version to illustrate the locations of these resistance mutations around the binding site.

COMMENT #8: Line 249 states that "the two most commonly observed resistance mutations" are C4657M/G4819E. However, in the introduction (line 105), it's mentioned that "I4790M and G4946E are the most common mutations found...", though in lines 278-279, it's noted that the residues are equivalent. This equivalence should be clarified at the beginning to avoid confusion and potential misinterpretation.

RESPONSE #8: We revised the text to clarify the equivalence between the residues chiRyR-I4657M/G4819E was first mentioned. The revised line 254-258 now explains that chiRyR-I4657M/G4819E corresponds to I4790M/G4946E in DBM RyR to avoid confusion and potential misinterpretation.

COMMENT #9: Line 256: there is no Supplementary Figure 7

RESPONSE #9: We have corrected this reference to Figure 4c in the revised version.

COMMENT #10: Line 264: the values reported in the text and the table data do not align

RESPONSE #10: We have corrected the values to ensure consistency between the text and the table data.

COMMENT #11: Line 282 and line 291: the cited figures do not correspond to the appropriate figure panels.

RESPONSE #11: We have corrected the figure references to ensure they correspond to the appropriate figure panels.

COMMENT #12: Line 326 and 328: The residues listed as LD₅₀ or LC₅₀ in Table 7 and the text are not consistent.

RESPONSE #12: Sorry about the confusion. We have rephrased the text to clarify the difference between Drosophila RyR numbering and Sf RyR numbering (lines 269-275): "To test whether this observation extends to insects, we used a gene-edited Drosophila melanogaster carrying the M4758I mutation (rendering it equivalent to wild-type Sf RyR). The LD₅₀ value for CHL for this transgene is 127.9 nM. Introducing the G4915E mutation to wild-type Drosophila melanogaster (the heterozygous mutation G4915E/G was introduced because the homozygous mutation is lethal), which makes it equivalent to the double-mutant Sf RyR-I4734M/G4891E, increases this value by four orders of magnitude (LD₅₀ = 1.6 mM) (Fig. 4d; Supplementary Table 6)."

COMMENT #13: The legend for Figure 2 does not match the panel labelings

RESPONSE #13: We have corrected the legend for Figure 2 to ensure it matches the panel labelings.

COMMENT #14: Figure S1: to be consistent with the figure legend, in panel a, FKBP12.6 needs to be shown or labeled in the right lane of the gel

RESPONSE #14: We have shown and labeled FKBP12.6 on the gel figure to ensure the consistency with the figure legend.

COMMENT #15: Line 214: what are 'important differences'?

RESPONSE #15: We have removed it in the revised version.

COMMENT #16: Line 370: should use 'among' instead of 'out of'?

RESPONSE #16: We have corrected it.

COMMENT #17: Figure 2a, it would be easier to compare the chemical structures if CHL was flipped 180 degrees to make the color box in the same order as TET

RESPONSE #17: We have revised the panel accordingly, flipping CHL 180 degrees to align the color box order with TET.

COMMENT #18: Consistency in color coding for maps and models across all figures would enhance convenience and ease of understanding.

RESPONSE #18: We have revised the color coding to ensure consistency across all figures.

COMMENT #19: In Figure 4, the label for E4819 should be more accurate; the last one appears blue, not red. For panel f, the purple arrow should be made more visible. In panel g, labels should be added for the two models.

RESPONSE #19: We thank the reviewer for the suggestion. We have revised them accordingly.

Reviewer #2 (Remarks to the Author):

Cryo-EM structures of Ryanodine Receptors and diamide insecticides reveal the mechanism for selectivity and resistance

The authors provide an analysis of the structure and function of chiRyR, a modified version of the rabbit ryanodine receptor type 1 protein, in the presence of two diamide insecticides, flubendiamide (FLU) and tetraniliprole (TET), highlighting the same binding site of FLU and TET in the pVSD domain of the transmembrane region of chiRyR. They also discuss the functional and structural impact of diamide resistance mutations in chiRyR, which decrease the effectiveness of FLU and CHL. The authors compare the diamide-binding pocket between the wild-type chiRyR and the chiRyR-C4657M/G4819E mutant and provide a schematic representation of chiRyR modulation by FLU, CHL, and the resistant mutations. Overall, the manuscript provides valuable insights into the molecular mechanisms underlying the binding and modulation of chiRyR by diamide insecticides, contributing to the understanding of pesticide resistance in insects and the development of more effective insecticides. However, there are major technical and analytical issues that need to be fixed and addressed.

COMMENT #1: The authors need to explain why recombinant expression of insect RyRs is challenging. They should also show the homology between insect RyR and the mammalian RyRs.

RESPONSE #1: To quantify the expression of *Sf* RyR, we performed the time-lapse experiments and showed that *Sf* RyR exhibits a similar level of drop in fluorescence signals, reflecting a comparable level of functional expression. However, when we attempted large-scale expression and purify using an FKBP-based affinity purification strategy, we found the final yield and purity are both quite low. This is likely due to the lower affinity between FKBP and insect RyR. Thus, in the future, we may need to focus on alternative purification strategies that still need to be established.

We have added the following explanation to the results section (lines 127-134): "Although mammalian RyRs can be obtained through recombinant expression and FKBP-based affinity purification²⁷⁻²⁸, insect RyRs have proven to be more challenging for larger-scale production and affinity purification because of their lower affinity for FKBP, which is needed for cryo-EM studies. The overall sequence identity between *Sf* RyR and rRyR1 is 45.2%, with higher identity (~60%) in their transmembrane regions, where the diamide binding site is located (Supplementary Fig. 1). Therefore, we designed a series of insect-like chimeric RyRs that contain the diamide-binding pocket from *Sf* RyR and the remaining parts from rabbit RyR1 (rRyR1)."

We also included a new supplementary figure of sequence alignment (Supplementary Figure 1) to illustrate the sequence identities between different domains of insect and mammalian RyRs.

COMMENT #2: Figure 2a should be figure 1a. and the structure of CHL should be flipped vertically so it's easier to compare to TET.

RESPONSE #2: We have moved Figure 2a to Figure 1a and flipped the structure of CHL vertically to align it with the orientation of TET for easier comparison.

COMMENT #3: In general, the figure panels should be better organized and ordered to correspond to the text.

RESPONSE #3: We have rearranged and optimized the organization of the figure panels to better correspond to the text.

COMMENT #4: The claimed local resolution of 3.2Å in figure S2 is not convincing since the scale goes from 3.2Å to 7.2Å. The subpanel should be focused on the ligand binding site and the scale should be from 3Å to 4Å to show that local resolution is approximately 3.2Å and not 3.82Å and 3.71Å as it is shown in figure S1e. Methodologically, the authors could improve the local resolution by expanding the particles on a C4 basis, creating a mask of one of the binding sites (one protomer only), and performing local refinement with the expanded particles and the local mask.

RESPONSE #4: We thank the reviewer for the suggestion. We performed the following steps based on the reviewer's recommendations:

1. Applied a mask covering both transmembrane domain (TMD) and C-terminal domain (CTD) of RyR and performed local refinement with C4 symmetry;
2. Applied a mask covering only TMD and performed local refinement with C4 symmetry.
3. Applied as mask covering a single protomer of TMD and performed local refinement with C1 symmetry followed by symmetry expansion.

The resolutions obtained from these three methods for chiRyR and chiRyR-I4657M/G4819E were:

Method 1: 3.68 Å and 3.28 Å

Method 2: 3.85 Å and 3.55 Å

Method 3: 4.04 Å and 3.79 Å

Method 1 showed the best results for both proteins. Therefore, we used this protocol to process all datasets and updated the Figures and Results sections accordingly. In the new Supplementary Figure 3, we added the insets to show the local resolution of the binding sites and changed the scale to from 3 Å to 4 Å. Please note that both ourselves and others have previously used symmetry expansion + local masking to improve the local resolution of RyR cryo-EM structures (Kobayashi, T; et al. 2022, Nature Communications; Haji-Ghassemi, O; et al. 2023, Science Advances; Nayak, A. R; et al. 2022, Elife.). Whereas this approach was successful in improving density of various regions in the cytosolic region, it was not successful for the transmembrane region, likely because an individual protomer within the TM region is too small.

COMMENT #5: Figure S3 should be part of the main manuscript since it is fundamental to understand the good fit of the atomic model, which it is not supported. The fit of the ligands with a resolution of 3.2-3.8Å is poor and figure S3 is not convincing. Looking at the cryo-EM maps, TET and FLU are well placed, but the conformation of the ligand “flexible” motives cannot be precisely determined. In the case of CHL, the fit is worse than the other two, suggesting that although it is clear that CHL is there, the pose of the ligand cannot be precisely determined. This needs to be address and any conclusions at the atomic level should be avoided or made with caution.

RESPONSE #5: As mentioned above, after reprocessing the datasets, we have improved the local resolutions for all five structures. The current local resolutions of the binding sites for the three complex structures, chiRyR-FLU, chiRyR-TET, and chiRyR-I4657M/G4819E-CHL, are 3.65 Å, 3.63 Å, and 3.37 Å, respectively. At a 5 sigma level, the ligands and the surrounding coordinating residues show a good fit into the densities, giving us more confidence in the reliability of the model. We have incorporated the updated figures into the revised manuscript (Figure 2b,c; Figure 5a; Supplementary Figure 5) to provide a clearer illustration of the model fitting.

COMMENT #6: The MM-GBSA analysis is based on an atomic model with low confidence. MM-GBSA analysis is useful when the atomic model is accurate. At a resolution of 3.2-3.8Å, neither the ligands or the chiRyR sidechains can be confidently modelled. Any conclusions coming from this analysis should be made with caution. Some results are contradictory like in the case of chiRyR-C4657M/G4819E-CHL. My recommendation is to remove the whole MM-GBSA analysis from the manuscript since it adds no real value.

RESPONSE #6: We agree with the reviewer and have removed the results from MM-GBSA analysis from the manuscript.

COMMENT #7: It is claimed that Lys4563 and Asp 4815 form an ionic interaction. They should show the distance between sidechains and the cryo-EM maps to show that the sidechains are confidently placed. Unconvincing at a 3.72Å resolution.

RESPONSE #7: We have added the distance in between the side chains Lys4563 and Asp 4815 and the cryo-EM maps of ref-chiRyR in updated Figure 3c. In the results, we revised the sentence to “In ref-chiRyR, Lys4563 and Asp4815 form a putative ionic interaction”.

COMMENT #8: Ref-chiRyR processing is incomplete. As shown in figure 3a, the pore is in an intermediate conformation, suggesting a mixture of open and closed channels, as highlighted in line 230. This is confirmed by the relative low intensity of the pore in the cryo-EM map, suggesting a mix of conformations. Therefore, all comparisons to ref-chiRyR are incorrect. It is inappropriate to use this structure as a reference. The authors need to perform local 3D variability or 3D classification with TMD mask in order to separate the closed from the open particles. Once separated the authors need to build a model for the closed state and a model for the open state. The authors should also perform local 3D variability or 3D classification with TMD mask of the other conditions to convince the readers that there are no mixed closed/open conformations.

RESPONSE #8: We thank the reviewer for pointing this out. Following the reviewer’s suggestion, we reprocessed the dataset using local 3D variability analysis (3DVA) and local 3D classification using three types of masks (as mentioned above). But both methods failed to separate the closed from the open particles. We further performed 3D classification without any mask and the classification yielded two classes, a main class containing 17,326 particles that now corresponds to open channels, and another one containing 2,179 particles that still seems to represent RyRs, possibly closed, but the number is too small to assess definitively. Because the number of particles in the second class is too few, we could not obtain another distinct class with only closed-state conformation. The closed-state ref-chiRyR structure was predicted by homology modeling using the closed-state rRyR1 as a template. Further structural comparison has been made with the open-state ref-chiRyR structure and the homology model of the closed-state ref-chiRyR.

COMMENT #9: The comparison in Figure 3b-f is poorly done. They compare the closed state of rRyR1 to the open state of chi-RyR1 in the presence of FLU and TET. There are too many variables changing. The changes in the helices positions are mostly due to the change from the closed to the open state. The authors should compare the open state of chi-RyR1 in the presence of FLU and TET to the open state of ref-chiRyR. The authors should also show how these helices change in wt-rRyR1 or ref-chiRyR going from the closed to the open state. Only then can the real impact of the ligands be assessed. It might be that the ligands stabilize the open state, without inducing conformational changes.

RESPONSE #9: We thank the reviewer for the suggestion. In the updated Figure 3, we now compare chi-RyR-FLU and chiRyR-TET with ref-chiRyR (solved in the open-state) as well as a homology model of closed-state ref-chiRyR based on closed-state rRyR1. We showed that the binding of the diamide insecticides causes conformational changes relative to open-state ref-chiRyR on top of the difference between the closed-state and the open-state ref-chiRyR. We also added a schematic cartoon (Figure 3f) and a supplementary movie (Supplementary Movie 1) to explain the impact of diamide binding on channel gating.

We included these findings in the results (lines 234-250): “Since ref-chiRyR is in the open-state, to understand the diamide-induced gating process, we also created a homology model of closed-state ref-chiRyR (ref-chiRyR-closed-HM) on the basis of the cryo-EM structure of closed-state rRyR1 solved in the presence of caffeine, ATP, and Ca²⁺ (PDB ID 5TAQ). Compared with the ref-chiRyR, the binding of additional diamide ligands, FLU or TET, expands the pocket. This subsequently causes a displacement of the S4-S5 linker, which moves S5 outward to relax the constriction of the helical bundle in the pore (Fig. 3d-f). On the other hand, the binding of these ligands induces an outward movement of helix S3. This pushes the S2-S3 linker connecting with the U-motif and indirectly transfers the conformational change to the C-terminal domain (CTD) of RyR, pulling open the channel by rotating the cytosolic end of the helix S6 (Fig. 3d-f). To accommodate the binding of the ligands, both the S4-S5 and S2-S3 linkers in chiRyR-FLU and chiRyR-TET undergo a larger displacement than the one between ref-chiRyR-closed-HM and ref-chiRyR (Supplementary Movie 1). Furthermore, FLU induces slightly greater displacement in the S4-S5 linker than TET does, probably due to the additional contact between Arg4824 from the linker and the sulfide amine moiety in FLU.”

COMMENT #10: In figure 4b, the mutant is called chiRyR-C4657M/G4819E but the residue shown is I4657. Since, the residue in chiRyR is I4657, because it was already mutated from rRyR1, the mutant should be named chiRyR-I4657M/G4819E.

RESPONSE #10: We have changed the mutant name to chiRyR-I4657M/G4819E throughout the manuscript.

COMMENT #11: In figure 4b, the distance measurement between poorly defined sidechains due to the resolution of 3.6Å is inappropriate. It is better to show the cryo-EM maps and show experimentally how the pocket looks smaller. It would be interesting to know how this pocket changes between the closed and open states.

RESPONSE #11: We thank the reviewer for the suggestion. We have replaced Figure 4B with a new supplementary Figure 8 showing a comparison between the maps of the binding sites from ref-chiRyR and chiRyR-I4657M/G4819E. This clearly demonstrates that the double mutations reduce the volume of the pocket. Since the cryo-EM structure of ref-chiRyR is in an open conformation while chiRyR-I4657M/G4819E is in a closed conformation, we also included a panel showing a simulated map of ref-chiRyR in the closed state using a homology model based on the closed-state structure of rRyR1. This shows that while the pocket undergoes a conformational change upon gating, the change is not as dramatic as the one induced by the resistance mutations.

COMMENT #12: In figure 4e, the authors claim a movement of the Glu4819 sidechain. The authors should show the cryo-EM maps to support this claim, otherwise is not convincing due to the resolution.

RESPONSE #12: We have addressed this by showing the density of the side chain of Glu4819 in the revised Figure 5a.

COMMENT #13: In figure 4f, the authors claim a movement of CHL. However, looking at the cryo-EM map, the CHL ligand does not fit the cryo-EM density well enough. Due to the poor fit, any conclusions made regarding the movement of such ligand are unconvincing.

RESPONSE #13: We reprocessed the data of chiRyR-I4657M/G4819E-CHL and improved its resolution by local refinement. After refinement, the CHL ligand shows a reliable fit into the density map as shown in the revised Figure 5a.

COMMENT #14: The text mentions figure S7 which does not exist.

RESPONSE #14: We have corrected this error, and the text now refers to the updated Fig 5e.

COMMENT #15: Figure 2f and 4f are hard to understand.

RESPONSE #15: We have revised these two panels to simplify the comparisons.

COMMENT #16: In all figures where a distance or interaction is shown, a label with the value of such distance should be included.

RESPONSE #16: We have added the distances of the interactions on the figures.

COMMENT #17: The caffeine concentration used in the time-lapse experiments is not mentioned. A representative curve of the experiment should be shown.

RESPONSE #17: We added the range of caffeine used in the time-lapse experiments in the method. We also included a new figure panel (Supplementary Figure 7a) to show the caffeine activation curves of different RyR constructs and Supplementary Table 4 also shows the corresponding EC50 values.

COMMENT #18: In line 483, it says "HEK293 cells stably expressing chiRyR" but in line 485 it says "cells were induced". Are the cells stably expressing or inducible?

RESPONSE #18: The chiRyR gene was integrated into the genome of HEK293 cells with an inducible promoter that should be induced by doxycycline during protein expression. We have revised the sentence to (lines 531-533) "chiRyR (rRyR1-R4563K+F4564Y+C4653I+F4792S) or chiRyR-I4657M/G4819E were integrated into HEK293 cells with an inducible promoter".

COMMENT #19: The purification method is different for chiRyR1 and the mutant. Why is that? No explanation is given.

RESPONSE #19: We followed the traditional RyR purification protocol, which includes an FKBP12.6-based affinity purification step and a gel-filtration purification step, to purify chiRyR. During this process, we found that we lose significant amount of protein during the concentration step before loading onto gel-filtration column. Therefore, for the purification of the new construct chiRyR-I4657M/G4819E, we replaced the gel-filtration purification step with anion exchange purification step, which improved the yield while maintaining the purity. We added the explanation in the method section (lines 552-553): "For chiRyR-I4657M/G4819E, an anion exchange purification step replaced the gel filtration purification step to reduce protein loss during concentration."

COMMENT #20: 5% of DMSO is used for the mutant chiRyR in presence of CHL. The authors claimed it is needed to reach 625uM CHL, however the experiment in figure 4a shows that 100uM is enough for inhibition, suggesting 100uM can be used and 2% DMSO can also be used. DMSO promotes the closing of the channel, so any conclusions regarding the state of the channel in 5% DMSO should be taken with caution and clarified in the manuscript.

RESPONSE #20: Although the EC50 value of CHL against chiRyR-I4657M/G4819E is 10.3 μ M in the time-lapse experiment, we collected some cryo-EM data using chiRyR-I4657M/G4819E incubated with 2% DMSO and 100 μ M CHL but failed to identify the density of the ligand in the structure. Thus, we had to increase the amount of DMSO and the concentration of CHL to solve the complex structure of chiRyR-I4657M/G4819E-CHL. The cryo-EM structure of chiRyR-I4657M/G4819E was determined in 2% DMSO but also showed a similar closed-state conformation. In contrast, ref-chiRyR, chiRyR-FLU, chiRyR-TET were all solved with 2% DMSO, which adopt open conformation. These results suggest likely the closed-state is induced by the resistance mutations rather than the high concentration of DMSO in our case. To clarify the potential impact of DMSO on the conclusion, we added the following sentence in the discussion (lines 403-404): "While DMSO was required for the cryo-EM experiments, we cannot entirely rule out the possibility that DMSO had an additional effect on the particle distribution."

COMMENT #21: What is the logic of using a base model (PDB 6M2W) for the chiRyR and a completely different model with lower resolution and no CaM1234 (PDB 5TAQ) for the mutant chiRyR? It is not hard to build the model of the closed state from the open state, and the comparison would be more consistent and scientifically correct.

RESPONSE #21: We remodeled the two structures of chiRyR-I4657M/G4819E using 6M2W as the base model to maintain consistency.

Reviewer #3 (Remarks to the Author):

COMMENT: This is a very well written manuscript elucidating critical aspects of Diamide insecticide binding in their molecular targets and the precise role of insecticide resistance mutations, by using advanced Cryo-EM methodology. The study is comprehensive and it will guide future studies in the field of insecticide resistance, including expansions in vectors of diseases (such as malaria vectors). The detailed analysis of the binding site among different active ingredients of the same (major) diamide insecticide class, provides extremely useful information and evidence for insect pest control applications and resistance management.

Some implications in medical applications relevant with RyRs are also nicely discussed in the manuscript. A possible limitation one might consider, the insect-like chimeric RyRs (containing diamide binding pocket from insect and the remaining part from rabbit). However, as clearly explained, this is not a real issue, as all relevant mechanistic aspects are relevant. The missing structure of a diamide insecticide bound to a closed RyR for wild type (but not resistant mutant) would be good to be added, if logistics for addition possible and realistic (timeframe and feasibility).

Finally, the session (line 298) Diamide-resistant mutations stabilise the closed state of RyR could be better explained, so that generalist can easily digest this important information (not very clear).

RESPONSE: We thank the reviewer for the positive comments of our study.

For the missing of the complex structure of closed RyR with diamide, please see our response for the 2nd comment of the Reviewer 1.

For the clarification of how the resistance mutations stabilize the closed state of RyR, please see our response for the 6th comment of the Reviewer 1.

Best regards,

Zhiguang Yuchi and Filip Van Petegem